# Synthesis, Structure and Cytotoxicity Testing of Novel 7-(4,5-Dihydro-1*H*-imidazol-2-yl)-2-aryl-6,7-dihydro-2*H*-imidazo[2,1-*c*][1,2,4]triazol-3(5*H*)-Imine Derivatives

**DOI:** 10.3390/molecules25245924

**Published:** 2020-12-14

**Authors:** Łukasz Balewski, Franciszek Sączewski, Patrick J. Bednarski, Lisa Wolff, Anna Nadworska, Maria Gdaniec, Anita Kornicka

**Affiliations:** 1Department of Chemical Technology of Drugs, Faculty of Pharmacy, Medical University of Gdańsk, Al. Gen. J. Hallera 107, 80-416 Gdańsk, Poland; saczew@gumed.edu.pl (F.S.); annanadworska@gumed.edu.pl (A.N.); anita.kornicka@gumed.edu.pl (A.K.); 2Department of Pharmaceutical and Medicinal Chemistry, Institute of Pharmacy, University of Greifswald, F.-L. Jahn Strasse 17, D-17489 Greifswald, Germany; bednarsk@uni-greifswald.de (P.J.B.); lisa.wolff@uni-greifswald.de (L.W.); 3Faculty of Chemistry, Adam Mickiewicz University, ul. Uniwersytetu Poznańskiego 8, 61-614 Poznań, Poland; magdan@amu.edu.pl

**Keywords:** imidazo[2,1-*c*][1,2,4]triazol-3(5*H*)-imines, amides, sulfonamides, ureas, thioureas, X-ray analysis, in vitro cytotoxic activity

## Abstract

The appropriate 1-arylhydrazinecarbonitriles **1a–c** are subjected to the reaction with 2-chloro-4,5-dihydro-1*H*-imidazole (**2**), yielding 7-(4,5-dihydro-1*H*-imidazol-2-yl)-2-aryl-6,7-dihydro-2*H*-imidazo[2,1-*c*][1,2,4]triazol-3(5*H*)-imines **3a–c**, which are subsequently converted into the corresponding amides **4a–e**, **8a–c**, sulfonamides **5a–n**, **9**, ureas **6a–I**, and thioureas **7a–d**. The structures of the newly prepared derivatives **3a–c**, **4a–e**, **5a–n**, **6a–i**, **7a–d**, **8a–c**, and **9** are confirmed by IR, NMR spectroscopic data, as well as single-crystal X-ray analyses of **5e** and **8c**. The in vitro cytotoxic potency of these compounds is determined on a panel of human cancer cell lines, and the relationships between structure and antitumor activity are discussed. The most active 4-chloro-*N*-(2-(4-chlorophenyl)-7-(4,5-dihydro-1*H*-imidazol-2-yl)-6,7-dihydro-2*H*-imidazo[2,1-*c*][1,2,4]triazol-3(5*H*)-ylidene)benzamide (**4e**) and *N*-(7-(4,5-dihydro-1*H*-imidazol-2-yl)-2-(*p*-tolyl)-6,7-dihydro-2*H*-imidazo[2,1-*c*][1,2,4]triazol-3(5*H*)-ylidene)-[1,1′-biphenyl]-4-sulfonamide (**5l**) inhibits the growth of the cervical cancer SISO and bladder cancer RT-112 cell lines with IC_50_ values in the range of 2.38–3.77 μM. Moreover, *N*-(7-(4,5-dihydro-1*H*-imidazol-2-yl)-2-phenyl-6,7-dihydro-2*H*-imidazo[2,1-*c*][1,2,4]triazol-3(5*H*)-ylidene)-4-phenoxybenzenesulfonamide (**5m**) has the best selectivity towards the SISO cell line and induces apoptosis in this cell line.

## 1. Introduction

Commonly used antineoplastic drugs represent a group of structurally diverse compounds. Their incomplete efficacy and the acquired resistance of tumor cells remain major challenges in cancer treatment. Both imidazoline [1,2] and triazole scaffolds display important fragments in several promising classes of compounds with an interesting pharmacological profile [3,4,5,6,7,8]. Special attention is paid to imidazolines with anticancer activity. One of the best-known imidazoline-containing topoisomerase inhibitors is bisantrene, used for many years in the treatment of certain types of leukemia [9,10]. Furamidazoline (DB60), another topoisomerase inhibitor, inhibits the growth of various tumor cell lines, including the cisplatin-resistant line [11]. Moreover, imidazoline derivatives have shown a strong ability to inhibit the growth of human cancer cell lines such as H460, HeLa, MiaPaCa-2, SW620, and MCF-7 in micromolar concentrations. The activity of the compounds is related to the influence on DNA and tubulin [12]. Nutlins, imidazoline-containing small-molecule inhibitors blocking the MDM2-p53 protein-protein interaction [13,14], have been advanced into early phase clinical trials (RG7112, RO5045337 (Figure 1), NCT01164033, NCT00623870, NCT005595533) [15]. 2-Aminoimidazolines are creatine kinase (CK) or creatine transport inhibitors, and have recently been patented as novel anticancer agents [16]. Water-soluble derivatives bearing the 2-aminoimidazoline moiety arrest tumor cells in the G2/M phase [17]. Moreover, imidazoline derivatives were identified as potent oligodendrocyte lineage transcription factor 2 (OLIG2) inhibitors, promising agents in the treatment of glioblastoma tumors [18,19]. Fused imidazoline derivative (*S*)-6-phenyl-2,3,5,6-tetrahydroimidazo[2,1-*b*]thiazole—levamisole (Figure 1), an anti-parasitic drug, is used with 5-fluorouracil in adjuvant therapy in patients after colorectal tumor surgery due to its immunostimulatory activity. Levamisole reduced the frequency of relapses and improved prognosis [20] and completed phase 3 clinical trials for colon cancer stage III treatment (NCT00309530). On the other hand, the 1,2,4-triazole scaffold is present in the structure of anticancer drugs such as anastrozole, vorozole, and letrozole [21]. Based on the idea of hybrid compounds [22,23,24,25,26,27,28,29], we reasoned that compounds incorporating both the imidazoline and triazole pharmacophore groups could be effective as chemotherapeutic agents. It should be pointed out that the imidazo-triazole moiety is a recurring motif of synthetic compounds of pharmacological interest [30,31]. The antiproliferative effects of the imidazo-triazole derivatives may result from inhibition of EPH-B3 and FGF-R1 tyrosine kinases [32]. In addition, imidazo-triazoles have been extensively explored due to their antimicrobial [30,33,34,35,36], antifungal [30,34], and antiviral activity [37].

As a part of our research aimed at finding new anticancer pharmacophore structures, we previously described the synthesis and pronounced anticancer activity of 2-imino-2*H*-chromen-3-yl-1,3,5-triazines, 3-(benzoxazol/benzothiazol-2-yl)-2*H*-chromen-2-imines, 8-chloro-5,5-dioxoimidazo[1,2-*b*][1,4,2]benzodithiazines, 2-amino-4-(3,5,5-trimethyl-2-pyrazolino)-1,3,5-triazines, copper(II) complexes of 2-substituted benzimidazoles, and *N*-(2-pyridyl)imidazolidin-2-ones(thiones) [28,29,38,39,40,41]. In this study, we chose to synthesize a small library of 7-(4,5-dihydro-1*H*-imidazol-2-yl)-2-aryl-6,7-dihydro-2*H*-imidazo[2,1-*c*][1,2,4]triazol-3(5*H*)-imine derivatives of types A–E (Figure 2) to identify compounds with potential antitumor activity.

## 2. Results and Discussions

### 2.1. Chemistry

Our research started with reactions of 1-arylhydrazinecarbonitriles **1a**–**c** [42] with 2-chloro-4,5-dihydro-1*H*-imidazole (**2**) [43]. As outlined in Scheme 1, the treatment of **1a**–**c** with an excess of 2-chloro-4,5-dihydro-1*H*-imidazole (**2**) in dichloromethane at ambient temperature yielded the desired 7-(4,5-dihydro-1*H*-imidazol-2-yl)-2-aryl-6,7-dihydro-2*H*-imidazo[2,1-*c*][1,2,4]triazol-3(5*H*)-imine derivatives **3a**–**c**.

The mechanism of the formation of **3a**–**c** may be explained as follows. The nucleophilic attack of the NH_2_ group of 1-arylhydrazinecarbonitriles **1a**–**c** at the carbon atom C-*2* of 2-chloro-4,5-dihydro-1*H*-imidazole (**2**) leads to the formation of the intermediate **A**. The use of a molar excess of 2-chloro-4,5-dihydro-1*H*-imidazole (**2**) allows the compound **A** to attack the second molecule of **2** to yield intermediate **B**. It should be noted that 2-chloroimidazoline (**2**) acts as a base in this process. In turn, the resulting intermediate **B** undergoes intramolecular cyclization to form fused imidazo-triazole derivatives **3a**–**c** (Scheme 1).

The imine moiety C=NH present in the structure of compounds **3a**–**c** allowed for their further transformations. Thus, reactions of **3a**–**c** carried out in chloroform with a variety of acyl chlorides or sulfonyl chlorides gave rise to the formation of the corresponding amides **4a**–**e** and sulfonamides **5a**–**n** in good yields (Scheme 2).

Upon treatment of 7-(4,5-dihydro-1*H*-imidazol-2-yl)-2-aryl-6,7-dihydro-2*H*-imidazo[2,1-*c*][1,2,4]triazol-3(5*H*)-imines **3a**–**c** with aryl isocyanates or aryl isothiocyanates, the corresponding urea **6a**–**i** and thiourea derivatives **7a**–**d** were formed (Scheme 3).

During the course of our experimental research, it was found that heating compounds **3a** and **3b** with a two-fold molar excess of acyl or sulfonyl chloride in the presence of triethylamine (TEA) leads to the formation of products substituted both at the nitrogen atom of the imine C=N-H moiety and at the *N*-1 position of the 4,5-dihydro-1*H*-imidazole ring. In this way, the corresponding *N*-(7-(1-benzoyl-4,5-dihydro-1*H*-imidazol-2-yl)-2-aryl-6,7-dihydro-2*H*-imidazo[2,1-*c*][1,2,4]triazol-3(5*H*)-ylidene)benzamides **8a–c** and *N*-(2-phenyl-7-(1-(phenylsulfonyl)-4,5-dihydro-1*H*-imidazol-2-yl)-6,7-dihydro-2*H*-imidazo[2,1-*c*][1,2,4]triazol-3(5*H*)-ylidene)benzenesulfonamide (**9**) were obtained (Scheme 4).

The structures of novel 7-(4,5-dihydro-1*H*-imidazol-2-yl)-2-aryl-6,7-dihydro-2*H*-imidazo[2,1-*c*][1,2,4]triazol-3(5*H*)-imine derivatives **3a**–**c**, **4a**–**e**, **5a**–**n**, **6a**–**i**, **7a**–**d**, **8a**–**c**, and **9** were confirmed by IR, NMR spectroscopic data (see NMR spectra in Appendix A), mass spectrometry, and elementary analysis. Thus, in the IR spectra of **3a**–**c**, bands in the range of 3408–3205 cm^−1^ are attributable to the N-H group, while the strong absorptions of the C=N group are observed in the range of 1531–1683 cm^−1^. In turn, the most diagnostic feature of the IR spectra of di-substituted derivatives **8a**–**c** and **9** is the absence of N-H bands.

In the ^1^H-NMR spectra of compounds **4**–**7**, a broad singlet corresponding to the proton of the N-H group of the imidazoline ring is present in the range of 5.45–6.33 ppm. The characteristic methylene protons CH_2_-CH_2_ of the fused imidazo-triazole moiety and 4,5-dihydro-1*H*-imidazole ring are found in the range of 3.38–4.79 ppm.

The ^13^C-NMR spectra recorded for 7-(4,5-dihydro-1*H*-imidazol-2-yl)-2-aryl-6,7-dihydro-2*H*-imidazo[2,1-*c*][1,2,4]triazol-3(5*H*)-imines **3**–**9** revealed three signals of quaternary carbon atoms: C_7a_=N, C_3_=N of fused imidazo-triazole, and C_2_=N of the 4,5-dihydro-1*H*-imidazole ring at 144, 150, and 155 ppm, respectively. The aliphatic carbons of 4,5-dihydro-1*H*-imidazole and imidazo-triazole moieties are found in the range of 40–53 ppm. The signals of the urea carbonyl group C=O of compounds **6a**–**i** are located in the range of 155.50–158.68 ppm. The ^13^C-NMR spectra of **7a** and **7d** showed signals at 181.54 and 182.08 ppm, which may be assigned to the quaternary carbon atom C=S of the thiourea group.

Moreover, the crystal structures of compounds **5e** and **8c** were determined by X-ray crystallography. The molecules of **5e** and **8c** contain a common 7-(4,5-dihydro-1*H*-imidazol-2-yl)-2,5,6,7-tetrahydro-3*H*-imidazo[2,1-*c*][1,2,4]triazol-3-imine fragment that adopts the same configuration in both molecules (Figure 3 and Figure 4). The amino N13 atom of the imidazolidine substituent shows a pyramidal arrangement of its bonds with the sum of valence angles equal to 343.4° in **5e** and 349.2° in **8c**. A weak intramolecular N13-H···N1 hydrogen-bond interaction with H···N1 distance of 2.4 Å is observed in **5e**. In turn, introduction of the acyl group at N13 results in an intramolecular strain that leads to a short contact of 2.822 Å between the imino N1 atom of the bicyclic system and the carbonyl C23 atom of the acyl group.

### 2.2. In Vitro Cytotoxic Activity

The in vitro cytotoxic potential of 7-(4,5-dihydro-1*H*-imidazol-2-yl)-2-aryl-6,7-dihydro-2*H*-imidazo[2,1-*c*][1,2,4]triazol-3(5*H*)-imine derivatives **3a**–**c**, **4a**–**e**, **5a**–**n**, **6a**–**i**, **7a**–**d**, **8a**–**c**, and **9** was evaluated against human cancer cell lines by the crystal violet microtiter plate assay as described earlier [44]. This assay measures the antiproliferative potency of compounds towards actively dividing cancer cells.

First, primary screening of the new compounds was done to indicate whether a compound possesses enough activity at a concentration of 10 μM or 20 μM to inhibit cell growth by 50%. The human tumor cell lines used were: human non-small cell lung cancer LCLC-103H, human cervix cancer SISO, human bladder carcinoma 5637, and human bladder carcinoma epithelial RT-112. Compounds that inhibited cell growth by more than 50% at 10 or 20 µM in one or more cell line were further investigated.

It should be noted that all the imines **3a**–**c** (Scheme 1), as well as the di-substituted amides **8a**–**c** and benzenesulfonamide **9** (Scheme 4) were inactive. On the other hand, for amide **4e**, sulfonamides **5e**, **5i**–**m**, ureas **6e**–**f**, and thiourea **7c**, which passed the preliminary test, a secondary screening to determine their potency was performed on two human tumor cell lines: human cervix cancer SISO and human bladder carcinoma epithelial RT-112. The results of the secondary screening are presented in Table 1 as the average IC_50_ values calculated from dose-response data after 96 h of exposure to the tested compounds.

In the series of amides **4a**–**e** (Scheme 2), only compound **4e** bearing electron-withdrawing groups R = Cl at position 4 of the phenyl ring and R^1^ = C_6_H_4_-Cl(*4*) of the amide functionality displayed growth inhibitory properties towards the two cell lines and showed slightly lower potency than the reference drug cisplatin (IC_50_ values 2.87–3.06 µM vs. 0.24–1.22 µM, Table 1). Other compounds with R = H, CH_3_ and R^1^ = C_6_H_5_, C_6_H_4_-CH_3_(*4*), C_6_H_4_-F(*4*), and C_6_H_4_-Cl(*4*) did not pass the preliminary test (**4a**–**d**, Scheme 2). This may suggest that the presence of two electron-withdrawing substituents at both R- and R^1^-positions is important for the inhibitory activity of the tested compounds.

For the sulfonamide series **5a**–**n** (Scheme 2), it was found that incorporation of a bulky lipophilic group at the R^2^-position of the sulfonamide moiety (R^2^ = 1-naphthyl, 2-naphthyl, C_6_H_4_-C_6_H_5_, or C_6_H_4_-O-C_6_H_5_) afforded compounds **5i**–**m** with good to high activity (IC_50_ = 2.38–8.13 µM, Table 1). The most active compound **5l** with methyl substituent at R-position (R = CH_3_) and 1,1′-biphenyl group at R^2^-position (R^2^ = C_6_H_4_-C_6_H_5_) displayed relatively high cell growth inhibitory potency (IC_50_ = 2.38–3.77 µM) compared to the reference cisplatin (IC_50_ = 0.24–1.96 µM). A slightly decreased antiproliferative activity was observed for unsubstituted at R-position analogue **5k** (R = H, R^2^ = C_6_H_4_-C_6_H_5_, IC_50_ = 3.42–5.59 µM). Likewise, when 1,1′-biphenyl at the R^2^-position in compound **5k** was replaced by the 1-naphthyl, 2-naphthyl, or phenoxyphenyl group, the resulting compounds **5i**, **5j**, and **5m** were less potent with antitumor activity limited to the SISO cell line (IC_50_ = 5.37–8.13 µM). Interestingly, the sulfonamide **5e** with R = H and R^2^ = C_6_H_4_-CH_3_(*4*) demonstrated moderate cytotoxic activity towards the SISO cell line (IC_50_ = 14.74 µM) while its analogue **5f** featuring the methyl group at the R-position (R = CH_3_, R^2^ = C_6_H_4_-CH_3_(*4*)), as well as other compounds with R = H, CH_3_ and R^2^ = CH_3_, C_6_H_5_ or a variously substituted phenyl ring did not pass the preliminary test (**5a**–**d**, **5g**–**h**, **5n**, Scheme 2). It could be suggested from these results that the combination of R = CH_3_ and R^2^ = C_6_H_4_-C_6_H_5_ results in a compound with optimal properties.

Similarly to amides **4a**–**e**, in the series of ureas **6a–i**, the best activity was found for compound **6f** with two electron-withdrawing substituents: R = Cl at position 4 of the phenyl ring and R^1^ = C_6_H_4_-Cl(*4*) of the urea moiety (IC_50_ = 3.75–6.01 µM, Table 1). Replacement of the Cl-substituent at the R-position for the electron-donating methyl group yielded less active analogue **6e** with selectivity to the cervical cancer cell line SISO (IC_50_ = 6.65 µM) over the bladder cancer cell line RT-112 (IC_50_ > 10 µM). Introduction of any of the substituents R = H, CH_3_ and R^1^ = C_6_H_5_, C_6_H_4_-CH_3_(*4*), and 1-naphthyl, SO_2_-C_6_H_4_-CH_3_(*4*), however, resulted in compounds that did not pass the preliminary test (**6a**–**d**, **6g**–**i**, Scheme 3).

In turn, among the thiourea derivatives **7a**–**d** (Scheme 3), the only substituent R^2^ of the thiourea moiety that produced moderate activity in the SISO cell line was C_6_H_4_-Cl(*4*) (**7c**: IC_50_ = 14.16 µM, Table 1).

The sulfonamide **5m**, which demonstrated pronounced cytotoxicity and selectivity for the cervical cancer cell line SISO (IC_50_ = 5.37 µM) over the bladder cancer cell line RT-112 (IC_50_ > 10 µM), was chosen to investigate whether it can induce apoptosis in the representative SISO cell line.

### 2.3. Induction of Apoptosis by Compound ***5m***

One of the most common methods used to detect apoptotic programmed cell death is to double stain treated cancer cells with the Annexin V-FITC (fluorescein isothiocyanate) and propidium iodide, which together distinguish cells as normal and in early or late stage of apoptosis. The Annexin V assay allows the quantification of the relative number of cancer cells undergoing apoptosis; by use of fluorescent flow cytometry the distribution of cells in early and late stages of apoptosis can be measured. In Figure 5 are summarized the average results of three independent experiments after treatment of the SISO cells for 24 h at the IC_50_ or doubled IC_50_ concentrations of compound **5m**. The fractions of early apoptotic cells are displayed on left y-axis and the late apoptotic cells on right y-axis. The left displays 24 h solvent control (DMF). After 24 h of treatment with the IC_50_ and the doubled IC_50_ of **5m**, 10.2 % and 17.2% of the SISO cells displayed signs of early apoptosis, respectively. As revealed, the percentage of cells in an early state of apoptosis increased with an increasing concentration of compound **5m**. In the case of late apoptotic cells, there was a significant increase when the IC_50_ concentration was doubled.

## 3. Experimental Section

### 3.1. Chemistry

The melting points were determined with a Boëtius apparatus and were uncorrected. The infrared spectra were recorded on a Nicolet 380 FT-IR spectrophotometer (Thermo Fisher Scientific, Waltham, MA, USA). Magnetic resonance spectra (NMR) (Agilent, Santa Clara, CA, USA) were recorded on a Varian Gemini 200 BB (200 MHz) spectrometer, a Varian Mercury-Vx300 spectrometer (300 MHz), and a Varian Unity Inova 500 (500 MHz) spectrometer in DMSO-d_6_ or CDCl_3_. The residual peaks of solvents were used as internal standards. Chemical shifts (*δ*) are given in ppm, and coupling constants (*J*) are given in Hz. Mass spectra were recorded on an LCMS 2010 spectrometer (Shimadzu, Tokyo, Japan). The compounds were identified based on their molecular ions obtained through electrospray ionization. Compounds were purified by the use of preparative chromatography. Thin-layer chromatography was performed on silica gel plates with fluorescence detection (Merck Silica Gel 254, Merck KGaA, Darmstadt, Germany). After, drying spots were detected under UV light (λ = 254 nm). The elemental analyses of carbon, hydrogen, and nitrogen determined for the compounds were within ±0.4% of the theoretical values.

#### 3.1.1. A General Procedure for the Preparation of Compounds **3a**–**c**

To a stirred solution of 2.5 g of 2-chloro-4,5-dihydro-1*H*-imidazole (25 mmol) (**2**) in dichloromethane (25–30 mL), five millimoles of the appropriate 1-arylhydrazinecarbonitrile **1a–c** were added. When the exothermic reaction subsided, the reaction mixture was stirred at room temperature for 12 h. The precipitate was filtered and washed with dichloromethane (**1a**, **1c**) or the oily residue was separated by decantation and washed with dichloromethane (**1b**). After drying, the resulting precipitate or oily residue was mixed with cooled water (15 mL) and filtered. The cooled filtrate was basified with 15 mL of a 20% potassium carbonate solution. The precipitate (**3b**, **3c**) was separated by suction, washed with a small amount of cooled water, and dried or the resulting oil (**3a**) was extracted with chloroform (4 × 20 mL). The combined organic extract was dried with anhydrous magnesium sulfate(VI), filtered, and concentrated under reduced pressure.

*7-(4,5-Dihydro-1H-imidazol-2-yl)-2-phenyl-6,7-dihydro-2H-imidazo[2,1-c][1,2,4]triazol-3(5H)-imine* (**3a**). Starting from 5 mmol (0.6658 g) of 1-phenylhydrazinecarbonitrile (**1a**), to the resulting oily residue, ten milliliters of anhydrous 2-propanone were added, and the precipitate was filtered and washed with a small amount of cooled 2-propanone. Compound **3a** was purified on silica gel by preparative thin-layer chromatography (chromatotron); eluent: ethyl acetate:methanol:triethylamine (7:2:1, *v*/*v*/*v*); yield 0,65 g (48%); m.p. 187–190 °C; IR (KBr, cm^−1^): 3404, 3315, 3205, 3065, 2946, 2862, 1677, 1626, 1533, 1287, 1217, 1055, 760; ^1^H-NMR (200 MHz, CDCl_3_): 3.66 (m, 4H, CH_2_-CH_2_), 3.99 (t, 2H, CH_2_), 4.39–4.47 (m, 2H, CH_2_), 5.05 (br.s, 2H, 2xNH), 7.15–7.22 (m, 1H, Ar), 7.30–7.44 (m, 2H, Ar), 7.53–7,57 (m, 2H, Ar); ^13^C-NMR (50 MHz, CDCl_3_): 39.88, 45.84, 51.04, 52.27, 121.17 (two overlapping signals), 125.72, 129.73 (two overlapping signals), 139.30, 149.84, 150.27, 155.51; *m*/*z* (ESI): 270 [M + H]^+^. Anal. Calcd for C_13_H_15_N_7_ (269.31): C, 57.98; H, 5.61; N, 36.41. Found: C, 57.91; H, 5.58; N, 36.11.

*7-(4,5-Dihydro-1H-imidazol-2-yl)-2-(p-tolyl)-6,7-dihydro-2H-imidazo[2,1-c][1,2,4]triazol-3(5H)-imine* (**3b**). Starting from 5 mmol (0.7359 g) of 1-(*p*-tolyl)hydrazinecarbonitrile (**1b**), compound **3b** was purified on silica gel by preparative thin-layer chromatography (chromatotron); eluent: ethyl acetate:methanol:triethylamine (8:1:1, *v*/*v*/*v*); crystallized from acetonitrile; yield 0,9 g (64%); m.p. 213–216 °C; IR (KBr, cm^−1^): 3408, 3361, 3319, 3240, 3206, 3065, 2948, 2887, 2863, 1676, 1625, 1592, 1531, 1500, 1451, 1390, 1331, 1287, 1253, 1055, 1025, 975, 759, 707; ^1^H-NMR (500 MHz, DMSO-*d*_6_): 2.26 (s, 3H, CH_3_), 3.48 (s, 4H, CH_2_-CH_2_), 3.85 (t, 2H, CH_2_), 4.24 (t, 2H, CH_2_), 5.80–6.20 (br.s, 2H, 2xNH), 7.14 (d, *J* = 8.8 Hz, 2H, Ar), 7.85 (d, *J* = 8.8 Hz, 2H, Ar); ^13^C-NMR (125 MHz, DMSO-*d*_6_+TFA): 21.13, 43.06, 44.34 (two overlapping signals), 53.66, 125.69 (two overlapping signals), 130.91 (two overlapping signals), 132.74, 140.75, 147.16, 149.68, 153.75; *m*/*z* (ESI): 284 [M + H]^+^. Anal. Calcd for C_14_H_17_N_7_ (283.33): C, 59.35; H, 6.05; N, 34.60. Found: C, 59.28; H, 5.99; N, 34.62.

*2-(4-Chlorophenyl)-7-(4,5-dihydro-1H-imidazol-2-yl)-6,7-dihydro-2H-imidazo[2,1-c][1,2,4]triazol-3(5H)-imine* (**3c**). Starting from 5 mmol (0.838 g) of 1-(4-chlorophenyl)hydrazinecarbonitrile (**1c**), compound **3c** was purified on silica gel by preparative thin-layer chromatography (chromatotron); yield 0.85 g (56%); m.p. 220–223 °C; IR (KBr, cm^−1^): 3384, 3243, 2963, 2930, 2892, 1683, 1625, 1594, 1537, 1492, 1390, 1289, 1255, 1063, 828; ^1^H-NMR (300 MHz, DMSO-*d*_6_): 3.51 (s, 4H, CH_2_-CH_2_), 3.87–3.89 (m, 2H, CH_2_), 4.24–4.28 (t, 2H, CH_2_), 6.16 (br.s, 2H, 2xNH), 7.37–7.39 (m, 2H, Ar), 8.12–8.14 (m, 2H, Ar); ^13^C-NMR (75 MHz, DMSO-*d*_6_): 45.64, 49.15, 51.21, 51.60, 118.74 (two overlapping signals), 125.97, 128.69 (two overlapping signals), 139.68, 149.10, 149.53, 155.12; *m*/*z* (ESI): 304 [M + H]^+^. Anal. Calcd for C_13_H_14_ClN_7_ (303.75): C, 51.40; H, 4.65; N, 32.28. Found: C, 51.36; H, 4.68; N, 32.34.

#### 3.1.2. A General Procedure for the Preparation of Compounds **4a**–**e** and **5a**–**n**

To a stirring solution of compound **3a**–**c** in anhydrous chloroform (5 mL), the appropriate aryl chloride or sulfonyl chloride (sulfonic acid chloride) was added (in the molar ratio of 1:1). The mixture was heated in an oil bath at 90 °C (compound **5m**: 20–22 °C) for 8–12 h. The progress of the reaction was controlled by TLC. After completion of the reaction, the mixture was evaporated under reduced pressure, and into the resulting residue, crushed ice was added. The precipitate was separated by suction and dried. The crude product was purified on silica gel by preparative thin-layer chromatography (chromatotron) or crystallization. In this manner, the following compounds were obtained.

*N-(7-(4,5-dihydro-1H-imidazol-2-yl)-2-phenyl-6,7-dihydro-2H-imidazo[2,1-c][1,2,4]triazol-3(5H)-ylidene)benzamide* (**4a**). Starting from 0.25 g (0.93 mmol) of **3a** and 0.131 g (0.108 mL, 0.93 mmol) of benzoyl chloride; yield 0.18 g (52%); eluent: dichloromethane:ethyl acetate:2-propanone:triethylamine (3:3:3:1, *v/v/v/v*); m.p. 219–222 °C; IR (KBr, cm^−1^): 3325, 3058, 2925, 2865, 1665, 1611, 1521, 1537, 1456, 1352, 1292, 721; ^1^H-NMR (200 MHz, CDCl_3_): 3.72–3.73 (m, 4H, CH_2_-CH_2_), 4.45–4.55 (m, 4H, CH_2_-CH_2_), 5.55 (br.s, 1H, NH), 7.28–7.36 (m, 1H, Ar), 7.41–7.49 (m, 5H, Ar), 8.02 (d, *J* = 7.9 Hz, 2H, Ar), 8.20 (d, *J* = 7.1 Hz, 2H, Ar); *m/z* (ESI): 374 [M + H]^+^. Anal. Calcd for C_20_H_19_N_7_O (373.41): C, 64.33; H, 5.13; N, 26.26. Found: C, 64.46; H, 5.06; N, 25.95.

*N-(7-(4,5-dihydro-1H-imidazol-2-yl)-2-(p-tolyl)-6,7-dihydro-2H-imidazo[2,1-c][1,2,4]triazol-3(5H)-ylidene)-4-methylbenzamide* (**4b**). Starting from 0.142g (0.5 mmol) of **3b** and 0.077 g (0.5 mmol) of p-toluoyl chloride; yield 0.1 g (25%); m.p. 238–240 °C; IR (KBr, cm^−1^): 3283, 3030, 2951, 2922, 2878, 1671, 1604, 1542, 1512, 1472, 1368, 1342, 1312, 1292, 815, 758; ^1^H-NMR (200 MHz, CDCl_3_): 2.39 (s, 6H, 2xCH_3_), 3.72 (br.s, 4H, CH_2_-CH_2_), 4.45–4.53 (m, 4H, CH_2_-CH_2_), 7.19–7.26 (m, 4H, Ar), 7.86 (d, *J* = 8.3 Hz, 2H, Ar), 8.08 (d, *J* = 7.9 Hz, 2H, Ar); 1H-NMR (200 MHz, CDCl_3_+TFA): 2.43 (s, 6H, 2xCH3), 4.08 (s, 4H, CH_2_-CH_2_), 4.94 (br.s, 4H, CH_2_-CH_2_), 7.28–7.37 (m, 6H, Ar), 7.07 (d, *J* = 7.9 Hz, 2H, Ar); 7.88 (br.s, 2H, NH+H^+^); *m/z* (ESI): 402 [M + H]^+^. Anal. Calcd for C_22_H_23_N_7_O (401.46): C, 65.82; H, 5.77; N, 24.42. Found: C, 65.78; H, 5.76; N, 24.38.

*N-(7-(4,5-dihydro-1H-imidazol-2-yl)-2-phenyl-6,7-dihydro-2H-imidazo[2,1-c][1,2,4]triazol-3(5H)-ylidene)-4-fluorobenzamide* (**4c**). Starting from 0.1347 g (0.5 mmol) of **3a** and 0.079 g (0.5 mmol) of 4-fluorobenzoyl chloride; yield 0.09 g (46%); m.p. 238–242 °C; IR (KBr, cm^−1^): 3392, 3072, 2930, 2876, 1675, 1618, 1599, 1517, 1457, 1286, 1217, 1146, 767; ^1^H-NMR (500 MHz, CDCl_3_): 3.74 (s, 4H, CH_2_-CH_2_), 4.47–4.52 (m, 2H, CH_2_), 4.54–4.58 (m, 2H, CH_2_), 7.06 (t, 2H, Ar), 7.30 (t, 1H, Ar); 7.45 (t, 2H, Ar); 7.97 (d, *J* = 7.8 Hz, 2H, Ar), 8.17–8.19 (m, 2H, Ar); ^13^C-NMR (125 MHz, CDCl_3_): 45.57 (two overlapping signals), 50.62 (two overlapping signals), 115.10 (d, *J*_(C-F)_ = 21.5 Hz, two overlapping signals), 122.01 (two overlapping signals), 126.84, 129.01 (two overlapping signals), 132.06 (d, *J*_(C-F)_ = 9.2 Hz, two overlapping signals), 133.63 (d, *J*_(C-F)_ = 3.1 Hz), 138.12, 148.37, 150.57, 154.85, 165.21 (d, *J*_(C-F)_ = 250.7 Hz), 171.60; *m/z* (ESI): 392 [M + H]^+^. Anal. Calcd for C_20_H_18_FN_7_O (391.40): C, 61.37; H, 4.64; N, 25.05. Found: C, 61.28; H, 4.60; N, 24.98.

*4-Chloro-N-(7-(4,5-dihydro-1H-imidazol-2-yl)-2-phenyl-6,7-dihydro-2H-imidazo[2,1-c][1,2,4]triazol-3(5H)-ylidene)benzamide* (**4d**). Starting from 0.20 g (0.74 mmol) of **3a** and 0.13 g (0.095 mL, 0.74 mmol) of 4-chlorobenzoyl chloride; yield 0.19 g (63%); eluent: dichloromethane:ethyl acetate:2-propanone:triethylamine (3:3:3:1, *v*/*v*/*v*/*v*); m.p. 249–251 °C; IR (KBr, cm^−1^): 3315, 3063, 2945, 2866, 1677, 1609, 1589, 1524, 1498, 1458, 1390, 1354, 1290, 1888, 1087, 1013, 760; ^1^H-NMR (200 MHz, CDCl_3_): 3.67–3.78 (m, 4H, CH_2_-CH_2_), 4.41–4.60 (m, 4H, CH_2_-CH_2_), 5.55 (br.s, 1H, NH), 7.24–7.50 (m, 5H, Ar), 7.96 (d, *J* = 7.6 Hz, 2H, Ar), 8.11 (d, *J* = 8.5 Hz, 2H, Ar); *m*/*z* (ESI): 408 [M + H]^+^. Anal. Calcd for C_20_H_18_ClN_7_O (407.86): C, 58.90; H, 4.45; N, 24.04. Found: C, 58.84; H, 4.42; N, 23.70.

*4-Chloro-N-(2-(4-chlorophenyl)-7-(4,5-dihydro-1H-imidazol-2-yl)-6,7-dihydro-2H-imidazo[2,1-c][1,2,4]triazol-3(5H)-ylidene)benzamide* (**4e**). Starting from 0.1519 g (0.5 mmol) of **3c** and 0.0875 g (0.064 mL, 0.5 mmol) of 4-chlorobenzoyl chloride; yield 0.12 g (54%); eluent: ethyl acetate:dichloromethane:methanol:triethylamine (6:2:1:1, *v/v/v/v*); crystallized from methanol; m.p. 278–283 °C; IR (KBr, cm^−1^): 3326, 3079, 3053, 3034, 2944, 2866, 1680, 1645, 1615, 1567, 1543, 1525, 1492, 1445, 1418, 1348, 1303, 1287, 1275, 1097, 1086, 1011, 831; ^1^H-NMR (400 MHz, DMSO-d_6_): 3.54 (br.s, 4H, CH_2_-CH_2_), 4.31–4.38 (m, 4H, CH_2_-CH_2_), 6.32 (br.s, 1H, NH), 7.51–7.61 (m, 4H, Ar), 8.07–8.16 (m, 4H, Ar); *m/z* (ESI): 442 and 444 [M + H]^+^. Anal. Calcd for C_20_H_17_Cl_2_N_7_O (442.30): C, 54.31; H, 3.87; N, 22.17. Found: C, 54.25; H, 3.92; N, 22.02.

*N-(7-(4,5-dihydro-1H-imidazol-2-yl)-2-phenyl-6,7-dihydro-2H-imidazo[2,1-c][1,2,4]triazol-3(5H)-ylidene)methanesulfonamide* (**5a**). Starting from 0.135 g (0.5 mmol) of **3a** and 0.0573 g (0.0387 mL, 0.5 mmol) of methanesulfonyl chloride; yield 0.1 g (58%); eluent: dichloromethane:ethyl acetate:2-propanone:triethylamine (3:3:3:1, *v/v/v/v*); m.p. 262–264 °C; IR (KBr, cm^−1^): 3378, 3072, 3015, 2950, 2875, 1681, 1643, 1606, 1578, 1521, 1498, 1460, 1271, 1128, 968, 934, 784, 762, 539; ^1^H-NMR (200 MHz, CDCl_3_): 3.10 (s, 3H, CH_3_), 3.69–3.74 (m, 4H, CH_2_-CH_2_), 4.45 (t, 2H, CH_2_), 4.66 (t, 2H, CH_2_), 5.49 (br.s, 1H, NH), 7.27 (t, 1H, Ar), 7.42 (t, 2H, Ar), 7.79 (d, *J* = 7.7 Hz, 2H, Ar); ^13^C-NMR (50 MHz, CDCl_3_): 43.43, 45.75 (br.), 46.75, 50.83, 53.00 (br.), 122.05 (two overlapping signals); 127.13, 129.26 (two overlapping signals), 137.95, 144.34, 155.45, 154.93; *m*/*z* (ESI): 348 [M + H]^+^. Anal. Calcd for C_14_H_17_N_7_O_2_S (347.40): C, 48.40; H, 4.93; N, 28.22. Found: C, 48.38; H, 4.89; N, 28.28.

*N-(7-(4,5-dihydro-1H-imidazol-2-yl)-2-(p-tolyl)-6,7-dihydro-2H-imidazo[2,1-c][1,2,4]triazol-3(5H)-ylidene)methanesulfonamide* (**5b**). Starting from 0.142 g (0.5 mmol) of **3b** and 0.0573 g (0.0387 mL, 0.5 mmol) of methanesulfonyl chloride; yield 0.1 g (55%); eluent: 2-propanone:ethyl acetate:dichloromethane:triethylamine (2:1:1:1, *v/v/v/v*); m.p. 258–262 °C; IR (KBr, cm^−1^): 3384, 3297, 3037, 3008, 2924, 2865, 1673, 1633, 1613, 1593, 1574, 1520, 1474, 1387, 1274, 1121, 968, 926, 782, 534; ^1^H-NMR (200 MHz, CDCl_3_): 2.37 (s, 3H, CH_3_), 3.08 (s, 3H, CH_3_), 3.71 (br.s, 4H, CH_2_-CH_2_), 4.45 (t, 2H, CH_2_), 4.66 (t, 2H, CH_2_), 5.45 (br.s, 1H, NH), 7.21 (d, *J* = 8.4 Hz, 2H, Ar); 7.64 (d, *J* = 8.4 Hz, 2H, Ar); ^13^C-NMR (50 MHz, CDCl_3_): 21.52, 43.42 (two overlapping signals), 45.71 (two overlapping signals), 50.81, 122.23 (two overlapping signals), 129.82 (two overlapping signals), 135.44, 137.23, 144.76, 150.36, 154.97; *m/z* (ESI): 362 [M + H]^+^. Anal. Calcd for C_15_H_19_N_7_O_2_S (361.42): C, 49.85; H, 5.30; N, 27.13. Found: C, 49.87; H, 5.30; N, 27.06.

*N-(7-(4,5-dihydro-1H-imidazol-2-yl)-2-phenyl-6,7-dihydro-2H-imidazo[2,1-c][1,2,4]triazol-3(5H)-ylidene)benzenesulfonamide* (**5c**). Starting from 0.18 g (0.668 mmol) of **3a** and 0.118 g (0.085 mL, 0.668 mmol) of benzenesulfonyl chloride; yield 0.18 g (66%); eluent: dichloromethane:ethyl acetate:2-propanone (1:2:2, *v/v/v*); crystallized from methanol; m.p. 233–234 °C; IR (KBr, cm^−1^): 3358, 3062, 2956, 2927, 2878, 1677, 1641, 1602, 1570, 1521, 1496, 1389, 1273, 1140, 1085, 925, 769, 690, 594; ^1^H-NMR (200 MHz, CDCl_3_): 3.51–3.91 (m, 4H, CH_2_-CH_2_), 4.46 (t, 2H, CH_2_), 4.72 (t, 2H, CH_2_), 5.49 (br.s, 1H, NH), 7.19–7.57 (m, 6H, Ar), 7.75 (d, *J* = 7.9 Hz, 2H, Ar), 7.94–7.99 (m, 2H, Ar); ^13^C-NMR (50 MHz, CDCl_3_): 45.83, 46.33, 50.84, 53.05, 121.97 (two overlapping signals), 126.59 (two overlapping signals), 127.16, 129,17 (two overlapping signals), 129.25 (two overlapping signals), 132.31, 137.88, 143.66, 144.32, 150.55, 154.92; *m/z* (ESI): 310 [M + H]^+^. Anal. Calcd for C_19_H_19_N_7_O_2_S (409.46): C, 55.73; H, 4.68; N, 23.95. Found: C, 55.69; H, 4.58; N, 24.30.

*N-(7-(4,5-dihydro-1H-imidazol-2-yl)-2-(p-tolyl)-6,7-dihydro-2H-imidazo[2,1-c][1,2,4]triazol-3(5H)-ylidene)benzenesulfonamide* (**5d**). Starting from 0.142 g (0.5 mmol) of **3b** and 0.118 g (0.085 mL, 0.668 mmol) of benzenesulfonyl chloride; yield 0.14 g (66%); eluent: ethyl acetate:2-propanone (1:1, v/v); crystallized from 2-propanone; m.p. 219–221 °C; IR (KBr, cm^−1^): 3361, 3104, 3058, 2953, 2870, 1667, 1638, 1591, 1568, 1513, 1445, 1386, 1314, 1277, 1145, 1087, 926, 760, 603; ^1^H-NMR (200 MHz, CDCl_3_): 2.33 (s, 3H, CH_3_), 3.56–3.72 (m, 2H, CH_2_), 3.73–3.81 (m, 2H, CH_2_), 4.45 (t, 2H, CH_2_), 4.71 (t, 2H, CH_2_), 5.49 (br.s, 1H, NH), 7.15 (d, *J* = 8.3 Hz, 2H, Ar), 7.42–7.50 (m, 3H, Ar), 7.60 (d, *J* = 8.3 Hz, 2H, Ar), 7.94–7.97 (m, 2H, Ar); ^13^C-NMR (50 MHz, CDCl_3_): 21.49, 45.81, 46.51 (br.), 50.83, 53.09 (br.), 122.08 (two overlapping signals), 126.59 (two overlapping signals), 129.13 (two overlapping signals), 129.80 (two overlapping signals), 132.22, 135.41, 137.20, 143.76, 144.22, 150.48, 154.92. *m/z* (ESI): 424 [M + H]^+^; Anal. Calcd for C_20_H_21_N_7_O_2_S (423.49): C, 56.72; H, 5.00; N, 23.15. Found: C, 56.68; H, 5.01; N, 23.12.

*N-(7-(4,5-dihydro-1H-imidazol-2-yl)-2-phenyl-6,7-dihydro-2H-imidazo[2,1-c][1,2,4]triazol-3(5H)-ylidene)-4-methylbenzenesulfonamide* (**5e**). Starting from 0.135 g (0.5 mmol) of **3a** and 0.095 g (0.5 mmol) of *p*-toluenesulfonyl chloride; yield 0.13 g (61%); crystallized from methanol; m.p. 227–231 °C; IR (KBr, cm^−1^): 3316, 3065, 2955, 2925, 2874, 1678, 1630, 1603, 1567, 1515, 1474, 1259, 1142, 1091, 926, 770, 565; ^1^H-NMR (200 MHz, CDCl_3_): 2.40 (s, 3H, CH_3_), 3.64–3.76 (m, 4H, CH_2_-CH_2_), 4.44 (t, 2H, CH_2_), 4.70 (t, 2H, CH_2_), 5.50 (br.s, 1H, NH), 7.18–7.29 (m, 5H, Ar), 7.74–7.86 (m, 4H, arom); ^13^C-NMR (50 MHz, CDCl_3_): 21.97, 45.85, 46.25 (br.), 50.82, 53.05 (br.), 121.88 (two overlapping signals), 126.61 (two overlapping signals), 127.02, 129.22 (two overlapping signals), 129.76 (two overlapping signals), 137.96, 140.90, 142.85, 144.29, 150.52, 154.91; *m/z* (ESI): 424 [M + H]^+^. Anal. Calcd for C_20_H_21_N_7_O_2_S (423.49): C, 56.72; H, 5.00; N, 23.15. Found: C, 56.69; H, 4.99; N, 23.27.

*N-(7-(4,5-dihydro-1H-imidazol-2-yl)-2-(p-tolyl)-6,7-dihydro-2H-imidazo[2,1-c][1,2,4]triazol-3(5H)-ylidene)-4-methylbenzenesulfonamide* (**5f**). Starting from 0.142 g (0.5 mmol) of **3b** and 0.095 g (0.5 mmol) of *p*-toluenesulfonyl chloride; yield 0.14 g (64%); eluent: 2-propanone:ethyl acetate (4:1, v/v); m.p. 203–206 °C; IR (KBr, cm^−1^): 3381, 3030, 2923, 2858, 1684, 1634, 1590, 1561, 1523, 1509, 1313, 1282, 1254, 1147, 1090, 935, 820, 556; ^1^H-NMR (200 MHz, CDCl_3_): 2.33 (s, 3H, CH_3_), 2.41 (s, 3H, CH_3_), 3.54–3.69 (m, 2H, CH_2_), 3.72–3.80 (m, 2H, CH_2_), 4.44 (t, 2H, CH_2_), 4.70 (t, 2H, CH_2_), 5.50 (br.s, 1H, NH), 7.15 (d, *J* = 8.4 Hz, 2H, Ar), 7.26 (d, *J* = 8.1 Hz, 2H, Ar), 7.61 (d, *J* = 8.4 Hz, 2H, Ar), 7.84 (d, *J* = 8.1 Hz, 2H, Ar); ^13^C-NMR (50 MHz, CDCl_3_): 21.48, 21.96, 45.82, 46.40 (br.), 50.82, 53.10 (br.), 122.01 (two overlapping signals), 126.61 (two overlapping signals), 129.72 (two overlapping signals), 129.78 (two overlapping signals), 135.48, 137.08, 140.97, 142.75, 144.20, 150.45, 154.95; *m/z* (ESI): 438 [M + H]+. Anal. Calcd for C_21_H_23_N_7_O_2_S (437.52): C, 57.65; H, 5.30; N, 22.41. Found: C, 57.61; H, 5.28; N, 22.44.

*N-(7-(4,5-dihydro-1H-imidazol-2-yl)-2-phenyl-6,7-dihydro-2H-imidazo[2,1-c][1,2,4]triazol-3(5H)-ylidene)-4-methoxybenzenesulfonamide* (**5g**). Starting from 0.269 g (1 mmol) of **3a** and 0.207 g (1 mmol) of 4-methoxybenzenesulfonyl chloride; yield 0.15 g (34%); eluent: chloroform:ethyl acetate:2-propanone:triethylamine (3:4:2:1, *v/v/v/v*); m.p. 207–209 °C; IR (KBr, cm^−1^): 3379, 3072, 2945, 2876, 1677, 1596, 1572, 1521, 1499, 1257, 1139, 1087, 769, 570; ^1^H-NMR (200 MHz, CDCl_3_): 3.71 (br.s, 4H, CH_2_-CH_2_), 3.86 (s, 3H, OCH_3_), 4.46 (t, 2H, CH_2_), 4.72 (t, 2H, CH_2_), 6.95 (d, *J* = 8.2 Hz, 2H, Ar), 7.19–7.40 (m, 3H, Ar), 7.76 (d, *J* = 7.7 Hz, 2H, Ar), 7.89 (d, *J* = 8.2 Hz, 2H, Ar); ^13^C-NMR (50 MHz, CDCl_3_): 45.34 (two overlapping signals), 50.32 (two overlapping signals), 55.50, 113.76 (two overlapping signals), 121.33 (two overlapping signals), 126.49, 128.11 (two overlapping signals), 128.70 (two overlapping signals), 135.16, 137.42, 143.70, 149.85, 154.38; 162,15; *m/z* (ESI): 440 [M + H]^+^. Anal. Calcd for C_20_H_21_N_7_O_3_S (439.49): C, 54.66; H, 4.82; N, 22.31. Found: C, 54.61; H, 4.74; N, 22.26.

*N-(7-(4,5-dihydro-1H-imidazol-2-yl)-2-phenyl-6,7-dihydro-2H-imidazo[2,1-c][1,2,4]triazol-3(5H)-ylidene)-4-nitrobenzenesulfonamide* (**5h**). Starting from 0.135 g (0.5 mmol) of **3a** and 0.111 g (0.5 mmol) of 4-nitrobenzenesulfonyl chloride; yield 0.19 g (84%); m.p. 263–269 °C; IR (KBr, cm^−1^): 3327, 3101, 3083, 2931, 2875, 1675, 1626, 1602, 1569, 1521, 1493, 1387, 1354, 1279, 1147, 1090, 926, 775, 747, 615; ^1^H-NMR (200 MHz, DMSO-d_6_): 3.52 (br.s, 4H, CH_2_-CH_2_), 4.34–4.36 (m, 4H, CH_2_-CH_2_), 6.33 (br.s, 1H, NH), 7.31–7.46 (m, 3H, Ar), 7.78 (d, *J* = 7.9 Hz, 2H, Ar), 8.1 (d, *J* = 8.3 Hz, 2H Ar), 8.37 (d, *J* = 8.7 Hz, 2H, Ar); *m/z* (ESI): 455 [M + H]^+^. Anal. Calcd for C_19_H_18_N_8_O_4_S (454.46): C, 50.21; H, 3.99; N, 24.66. Found: C, 50.19; H, 4.04; N, 24.78.

*N-(7-(4,5-dihydro-1H-imidazol-2-yl)-2-phenyl-6,7-dihydro-2H-imidazo[2,1-c][1,2,4]triazol-3(5H)-ylidene)naphthalene-1-sulfonamide* (**5i**). Starting from 0.134 g (0.5 mmol) of **3a** and 0.113 g (0.5 mmol) of 1-naphthalenesulfonyl chloride; yield 0.1 g (44%); eluent: dichloromethane:ethyl acetate:2-propanone:triethylamine:methanol (6:6:5:2:1, *v/v/v/v/v*); m.p. > 350 °C; IR (KBr, cm^−1^): 3371, 3058, 2928, 2878, 1672, 1632, 1590, 1566, 1518, 1388, 1263, 1112, 922, 769, 597, 509; ^1^H-NMR (200 MHz, CDCl_3_): 3.71 (s, 4H, CH_2_-CH_2_), 4.49 (t, 2H, CH_2_), 4.79 (t, 2H, CH_2_), 7.10–7.27 (m, 4H, Ar+NH), 7.48 (d, *J* = 7.5 Hz, 1H, Ar), 7.54–7.65 (m, 2H, Ar), 7.68 (d, *J* = 7.9 Hz, 2H, Ar), 7.89–7.94 (m, 1H, Ar), 8.02 (d, *J* = 7.9 Hz, 1H, Ar), 8.29 (d, *J* = 7.5 Hz, 1H, Ar), 8.81–8.86 (m, 1H, Ar); ^13^C-NMR (50 MHz, CDCl_3_): 45.79 (two overlapping signals), 50.63 (two overlapping signals), 121.73 (two overlapping signals), 124.30, 126.23, 126.85, 126.93, 127.69, 128.68, 128.79, 128.89 (three overlapping signals), 133.62, 134.48, 137.44, 138.56, 143.88, 150.29, 154.64; *m/z* (ESI): 460 [M + H]^+^. Anal. Calcd for C_23_H_21_N_7_O_2_S (459.52): C, 60.12; H, 4.61; N, 21.34. Found: C, 60.02; H, 4.55; N, 21.30.

*N-(7-(4,5-dihydro-1H-imidazol-2-yl)-2-phenyl-6,7-dihydro-2H-imidazo[2,1-c][1,2,4]triazol-3(5H)-ylidene)naphthalene-2-sulfonamide* (**5j**). Starting from 0.269 g (1 mmol) of **3a** and 0.227 g (1 mmol) of 2-naphthalenesulfonyl chloride; yield 0.13 g (28%); eluent: dichloromethane:ethyl acetate:2-propanone:triethylamine (1:3:3:3, *v/v/v/v*); crystallized from methanol; m.p. 214–217 °C; IR (KBr, cm^−1^): 3374, 3054, 2951, 2879, 1680, 1607, 1570, 1519, 1479, 1459, 1278, 1255, 1122, 1075, 925, 769, 751, 663, 558; ^1^H-NMR (200 MHz, CDCl_3_): 3.67–3.77 (m, 4H, CH_2_-CH_2_), 4.49 (t, 2H, CH_2_), 4.78 (t, 2H, CH_2_), 5.52 (br.s, 1H, NH), 7.23–7.35 (m, 3H, Ar), 7.60–7.62 (m, 2H, Ar), 7.78 (d, *J* = 7.5 Hz, 2H, Ar), 7.96 (m, 4H, Ar), 8.52 (s, 1H, Ar); ^13^C-NMR (50 MHz, CDCl_3_): 45.31, 45.87, 50.34, 52.54, 121.37 (two overlapping signals), 122.34, 126.56 (two overlapping signals), 127.12, 127.76, 128.21, 128.71 (two overlapping signals), 128.92, 129.18, 132.04, 134.44, 137.35, 140.01, 143.73, 150.01, 154.32; *m/z* (ESI): 460 [M + H]^+^. Anal. Calcd for C_23_H_21_N_7_O_2_S (459.52): C, 60.12; H, 4.61; N, 21.34. Found: C, 60.09; H, 4.58; N, 21.28.

*N-(7-(4,5-dihydro-1H-imidazol-2-yl)-2-phenyl-6,7-dihydro-2H-imidazo[2,1-c][1,2,4]triazol-3(5H)-ylidene)-[1,1’-biphenyl]-4-sulfonamide* (**5k**). Starting from 0.135 g (0.5 mmol) of **3a** and 0.126 g (0.5 mmol) of biphenyl-4-sulfonyl chloride; yield 0.12 g (49%); eluent: dichloromethane:ethyl acetate:2-propanone:triethylamine:methanol (4:3:1:1:1, *v/v/v/v/v*) or ethyl acetate:triethylamine (4:1, v/v); crystallized from methanol; m.p. 241–244 °C; IR (KBr, cm^−1^): 3370, 3058, 3033, 2965, 2923, 2823, 1673, 1631, 1605, 1575, 1511, 1258, 1139, 1093, 1033, 935, 766, 597; ^1^H-NMR (500 MHz, CDCl_3_): 3.64–3.79 (m, 4H, CH_2_-CH_2_), 4.49 (t, 2H, CH_2_), 4.74 (t, 2H, CH_2_), 5.52 (br.s, 1H, NH), 7.23–7.27 (m, 1H, Ar), 7.37 (t, 2H, Ar), 7.41 (d, *J* = 7.8 Hz, 1H Ar), 7.47 (t, 2H, Ar), 7.61 (d, *J* = 7.3 Hz, 2H, Ar), 7.69 (d, *J* = 8.3 Hz, 2H, Ar), 7.77 (d, *J* = 8.8 Hz, 2H, Ar), 8.02 (d, *J* = 8.8 Hz, 2H, Ar); ^13^C-NMR (125 MHz, CDCl_3_): 45.61 (two overlapping signals), 50.62 (two overlapping signals), 121.79 (two overlapping signals), 126.89 (two overlapping signals), 126.97, 127.55 (two overlapping signals), 127.62 (two overlapping signals), 128.46, 129.05 (two overlapping signals), 129.21 (two overlapping signals), 137.62, 139.91, 142.06, 144.07, 144.97, 150.32, 154.67; *m/z* (ESI): 486 [M + H]^+^. Anal. Calcd for C_25_H_23_N_7_O_2_S (485.56): C, 61.84; H, 4.77; N, 20.19. Found: C, 61.80; H, 4.71; N, 20.08.

*N-(7-(4,5-dihydro-1H-imidazol-2-yl)-2-(p-tolyl)-6,7-dihydro-2H-imidazo[2,1-c][1,2,4]triazol-3(5H)-ylidene)-[1,1’-biphenyl]-4-sulfonamide* (**5l**). Starting from 0.142 g (0.5 mmol) of **3b** and 0.126 g (0.5 mmol) of biphenyl-4-sulfonyl chloride; yield 0.09 g (36%); m.p. 214–218 °C; IR (KBr, cm^−1^): 3360, 3069, 3034, 2980, 2950, 2874, 1670, 1638, 1591, 1570, 1517, 1450, 1388, 1281, 1144, 1090, 924, 672, 604; ^1^H-NMR (300 MHz, CDCl_3_): 2.33 (s, 3H, CH_3_), 3.71 (br.s, 4H, CH_2_-CH_2_), 4.48 (t, 2H, CH_2_), 4.74 (t, 2H, CH_2_), 5.47 (br.s, 1H, NH), 7.17 (d, *J* = 8.4 Hz, 2H, Ar), 7.39–7.49 (m, 3H, Ar); 7.59–7.70 (m, 6H, Ar), 7.99–8.03 (m, 2H, Ar); ^13^C-NMR (75 MHz, CDCl_3_): 21.01, 45.34, 46.28, 50.37, 52.14, 121.63 (two overlapping signals), 126.62 (three overlapping signals), 127.29, 127.32 (two overlapping signals), 128.17, 128.94, 128.95, 129.34, 129.38, 134.89, 136.77, 139.70, 141.90, 143.71, 144.64, 149.97, 154.43; *m/z* (ESI): 500 [M + H]^+^. Anal. Calcd for C_26_H_25_N_7_O_2_S (499.59): C, 62.51; H, 5.04; N, 19.63. Found: C, 62.55; H, 4.89; N, 19.58.

*N-(7-(4,5-dihydro-1H-imidazol-2-yl)-2-phenyl-6,7-dihydro-2H-imidazo[2,1-c][1,2,4]triazol-3(5H)-ylidene)-4-phenoxybenzenesulfonamide* (**5m**). Starting from 0.135 g (0.5 mmol) of **3a** and 0.134 g (0.5 mmol) of 4-phenoxybenzenesulfonyl chloride; yield 0.08 g (32%); eluent: ethyl acetate:methanol (9:1, v/v and 8:2, v/v); crystallized from methanol; m.p. 186–188 °C; IR (KBr, cm^−1^): 3372, 3247, 3069, 2924, 2848, 1671, 1607, 1590, 1522, 1488, 1390, 1280, 1266, 1241, 1142, 1087, 927, 769, 578; ^1^H-NMR (500 MHz, DMSO-d_6_): 3.46–3.55 (m, 4H, CH_2_-CH_2_), 4.29 (t, 2H, CH_2_), 4.40 (t, 2H, CH_2_), 6.11 (s, 1H, NH), 7.07 (d, *J* = 8.8 Hz, 2H, Ar), 7.12 (d, *J* = 7.8 Hz, 2H, Ar), 7.23 (t, 1H, Ar), 7.28 (t, 1H, Ar), 7.42 (d, *J* = 7.3 Hz, 2H, Ar), 7.45 (d, *J* = 7.3 Hz, 2H, Ar), 7.80 (d, *J* = 7.8 Hz, 2H, Ar), 7.84 (d, *J* = 8.8 Hz, 2H, Ar); ^13^C-NMR (125 MHz, DMSO-d_6_+TFA): 44.30 (two overlapping signals), 46.49, 52.61, 118.18 (two overlapping signals), 120.64 (two overlapping signals), 122.34 (two overlapping signals), 125.43, 127.76, 128.76 (two overlapping signals), 129.64 (two overlapping signals), 130.98 (two overlapping signals), 137.72, 138.35, 144.21, 149.57, 153.89, 155.67, 160.69; *m/z* (ESI): 502 [M + H]^+^. Anal. Calcd for C_25_H_23_N_7_O_3_S (501.56): C, 59.87; H, 4.62; N, 19.55. Found: C, 59.87; H, 4.62; N, 19.55.

*N-(7-(4,5-dihydro-1H-imidazol-2-yl)-2-phenyl-6,7-dihydro-2H-imidazo[2,1-c][1,2,4]triazol-3(5H)-ylidene)-4-(trifluoromethyl)benzenesulfonamide* (**5n**). Starting from 0.135 g (0.5 mmol) of **3a** and 0.122 g (0.5 mmol) of 4-(trifluoromethyl)benzenesulfonyl chloride; yield 0.1 g (42%); eluent: dichloromethane:methanol (9:1, v/v); m.p. 221–223 °C; IR (KBr, cm^−1^): 3379, 3076, 2933, 2876, 1677, 1605, 1574, 1522, 1499, 1471, 1458, 1389, 1324, 1274, 1168, 1143, 1093, 1063, 939, 926, 769, 724, 610; ^1^H-NMR (500 MHz, DMSO-d_6_): 3.51 (s, 4H, CH_2_-CH_2_), 4.29–4.32 (m, 2H, CH_2_), 4.39–4.42 (m, 2H, CH_2_), 6.18 (br.s, 1H, NH), 7.31 (t, 1H, Ar), 7.45 (t, 2H, Ar), 7.77–7.79 (m, 2H, Ar), 7.93 (d, *J* = 8.3 Hz, 2H, Ar), 8.07 (d, *J* = 8.3 Hz, 2H, Ar); ^13^C-NMR (75 MHz, DMSO-d_6_): 45.58 (two overlapping signals), 51.21 (two overlapping signals), 122.21 (two overlapping signals), 124.03 (q, ^1^*J*_(C-F)_ = 272 Hz), 126.71 (q, ^3^*J*_(C-F)_ = 4.3 Hz, two overlapping signals), 127.00 (two overlapping signals), 127.25, 129.31 (two overlapping signals), 131.54 (q, ^2^*J*_(C-F)_ = 32 Hz), 137.59, 143.90, 148.17, 151.04, 154.74; *m/z* (ESI): 478 [M + H]^+^. Anal. Calcd for C_20_H_18_F_3_N_7_O_2_S (477.46): C, 50.31; H, 3.80; N, 20.53. Found: C, 50.26; H, 3.84; N, 20.47.

#### 3.1.3. A General Procedure for the Preparation of Compounds **6a**–**i** and **7a**–**d**

To a stirring solution of compound **3a–c** in anhydrous dichloromethane (5 mL), the appropriate aryl isocyanate or isothiocyanate was added (in the molar ratio of 1:1). The mixture was stirred at room temperature (20–22 °C) for 12 h. The progress of the reaction was controlled by TLC. After completion of the reaction, the precipitate was separated by suction, washed with a small amount of dichloromethane, and dried. The crude product was purified on silica gel by preparative thin-layer chromatography (chromatotron) or crystallization. In this manner, the following compounds were obtained.

*1-(7-(4,5-Dihydro-1H-imidazol-2-yl)-2-phenyl-6,7-dihydro-2H-imidazo[2,1-c][1,2,4]triazol-3(5H)-ylidene)-3-phenylurea* (**6a**). Starting from 0.135 g (0.5 mmol) of **3a** and 0.0596 g (0.0543 mL, 0.5 mmol) of phenyl isocyanate; yield 0.12 g (62%); eluent: chloroform:ethyl acetate:2-propanone:methanol (1:1:1:1, *v/v/v/v*); m.p. 241–247 °C; IR (KBr, cm^−1^): 3402, 3201, 3076, 2930, 2873, 1686, 1633, 1598, 1570, 1512, 1497, 1436, 1305, 1234, 1143, 748; ^1^H-NMR (500 MHz, DMSO-d_6_): 3.52 (s, 4H, CH_2_-CH_2_), 4.25 (t, 2H, CH_2_), 4.34 (t, 2H, CH_2_), 6.21 (br.s, 1H, NH), 6.88 (t, 1H, Ar), 7.19–7.24 (m, 3H, Ar), 7.44 (t, 2H, Ar), 7.61 (d, *J* = 7.8 Hz, 2H, Ar), 8.11 (d, *J* = 7.8 Hz, 2H, Ar), 9.17 (s, 1H, NH); ^1^H-NMR (500 MHz, DMSO-d_6_+TFA): 3.85 (s, 4H, CH_2_-CH_2_), 4.41–4.48 (m, 4H, CH_2_-CH_2_), 6.90 (t, 1H, Ar), 7.22 (t, 2H, Ar), 7.28 (t, 1H, Ar), 7.48 (t, 2H, Ar), 7.62 (d, *J* = 7.3 Hz, 2H, Ar), 8.18 (d, *J* = 8.3 Hz, 2H, Ar), 9.20 (br.s, 2H, NH+NH^+^), 9.31 (s, 1H, NH); ^13^C-NMR (125 MHz, DMSO-d_6_+TFA): 44.29 (two overlapping signals), 46.20, 52.20, 118.86 (two overlapping signals), 121.54 (two overlapping signals), 122.12, 126.44, 129.10 (two overlapping signals), 129.44 (two overlapping signals), 138.93, 141.46, 146.90, 149.28, 154.00, 158.33; *m/z* (ESI): 389 [M + H]^+^. Anal. Calcd for C_20_H_20_N_8_O (388.43): C, 61.84; H, 5.19; N, 28.85. Found: C, 61.80; H, 5.15; N, 28.79.

*1-(7-(4,5-Dihydro-1H-imidazol-2-yl)-2-(p-tolyl)-6,7-dihydro-2H-imidazo[2,1-c][1,2,4]triazol-3(5H)-ylidene)-3-phenylurea* (**6b**). Starting from 0.142 g (0.5 mmol) of **3b** and 0.0596 g (0.0543 mL, 0.5 mmol) of phenyl isocyanate; yield 0.16 g (80%); eluent: chloroform:ethyl acetate:2-propanone:methanol (1:1.5:1:0.5, *v/v/v/v*); crystallized from chloroform:ethyl acetate:methanol (1:1:1, *v/v/v*); m.p. 242–246 °C; IR (KBr, cm^−1^): 3411, 3312, 3046, 2950, 2873, 1684, 1638, 1589, 1570, 1498, 1432, 1303, 1223, 1141, 820, 758; ^1^H-NMR (500 MHz, DMSO-d_6_): 2.32 (s, 3H, CH_3_), 3.52 (s, 4H, CH_2_-CH_2_), 4.24 (t, 2H, CH_2_), 4.33 (t, 2H, CH_2_), 6.25 (br.s, 1H, NH), 6.87 (t, 1H, Ar), 7.20 (t, 2H, Ar), 7.24 (d, *J* = 8.3 Hz, 2H, Ar), 7.60 (d, *J* = 8.3 Hz, 2H, Ar), 7.96 (d, *J* = 8.3 Hz, 2H, Ar), 9.12 (s, 1H, NH); ^13^C-NMR (125 MHz, DMSO-d_6_+TFA): 20.87, 44.21 (two overlapping signals), 46.75, 52.28, 119.18 (two overlapping signals), 122.73, 123.18 (two overlapping signals), 129.09 (two overlapping signals), 130.05 (two overlapping signals), 135.18, 137.85, 140.41, 145.83, 149.67, 153.90, 155.50; *m/z* (ESI): 403 [M + H]^+^. Anal. Calcd for C_21_H_22_N_8_O (402.45): C, 62.67; H, 5.51; N, 27.84. Found: C, 62.65; H, 5.55; N, 27.92.

*1-(7-(4,5-Dihydro-1H-imidazol-2-yl)-2-phenyl-6,7-dihydro-2H-imidazo[2,1-c][1,2,4]triazol-3(5H)-ylidene)-3-(p-tolyl)urea* (**6c**). Starting from 0.135 g (0.5 mmol) of **3a** and 0.0665 g (0.063 mL, 0.5 mmol) of *p*-tolyl isocyanate; crystallized from methanol; yield 0.16 g (80%); m.p. 262–264 °C; IR (KBr, cm^−1^): 3388, 3207, 3062, 3026, 2948, 2877, 1691, 1638, 1604, 1588, 1573, 1513, 1403, 1313, 1291, 1235, 1145, 769, 748; ^1^H-NMR (500 MHz, DMSO-d_6_): 2.21 (s, 3H, CH_3_), 3.52 (s, 4H, CH_2_-CH_2_), 4.25 (t, 2H, CH_2_), 4.33 (t, 2H, CH_2_), 6.20 (br.s, 1H, NH), 7.01 (d, *J* = 8.3 Hz, 2H, Ar), 7.22 (t, 1H, Ar), 7.43 (t, 2H, Ar), 7.49 (d, *J* = 7.3 Hz, 2H, Ar), 8.11 (d, *J* = 7.8 Hz, 2H, Ar), 9.08 (s, 1H, NH); ^13^C-NMR (125 MHz, DMSO-d_6_+TFA): 20.57, 44.20 (two overlapping signals), 46.71, 52.27, 119.25 (two overlapping signals), 122.95 (two overlapping signals), 127.48, 129.48 (two overlapping signals), 129.58 (two overlapping signals), 131.76, 137.79, 137.85, 145.88, 149.71, 153.92, 155.63; *m/z* (ESI): 403 [M + H]^+^. Anal. Calcd for C_21_H_22_N_8_O (402.45): C, 62.67; H, 5.51; N, 27.84. Found: C, 62.67; H, 5.51; N, 27.84.

*1-(7-(4,5-Dihydro-1H-imidazol-2-yl)-2-(p-tolyl)-6,7-dihydro-2H-imidazo[2,1-c][1,2,4]triazol-3(5H)-ylidene)-3-(p-tolyl)urea* (**6d**). Starting from 0.142 g (0.5 mmol) of **3b** and 0.0665 g (0.063 mL, 0.5 mmol) of *p*-tolyl isocyanate; eluent: ethyl acetate:2-propanone:methanol (7:1:2, *v/v/v*); crystallized from methanol; yield 0.11 g (53%); m.p. 258–262 °C; IR (KBr, cm^−1^): 3398, 3205, 3079, 3030, 2958, 2919, 2876, 1686, 1637, 1587, 1514, 1312, 1291, 1236, 1181, 1144, 813; ^1^H-NMR (500 MHz, DMSO-d_6_): 2.20 (s, 3H, CH_3_), 2.32 (s, 3H, CH_3_), 3.52 (s, 4H, CH_2_-CH_2_), 4.24 (t, 2H, CH_2_), 4.33 (t, 2H, CH_2_), 6.25 (br.s, 1H, NH), 7.00 (d, *J* = 8.3 Hz, 2H, Ar), 7.23 (d, *J* = 8.3 Hz, 2H, Ar), 7.48 (d, *J* = 7.8 Hz, 2H, Ar), 7.96 (d, *J* = 8.3 Hz, 2H, Ar), 9.03 (s, 1H, NH); ^13^C-NMR (125 MHz, DMSO-d_6_+TFA): 20.58, 20.88, 44.23 (two overlapping signals), 47.12, 52.39, 119.38 (two overlapping signals), 123.92 (two overlapping signals), 129.59 (two overlapping signals), 130.25 (two overlapping signals), 132.25, 134.52, 137.25, 138.79, 145.25, 150.01, 153.90, 158.64; *m/z* (ESI): 417 [M + H]^+^. Anal. Calcd for C_22_H_24_N_8_O (416.48): C, 63.45; H, 5.81; N, 26.90. Found: C, 63.48; H, 5.74; N, 26.86.

*1-(4-Chlorophenyl)-3-(7-(4,5-dihydro-1H-imidazol-2-yl)-2-(p-tolyl)-6,7-dihydro-2H-imidazo[2,1-c][1,2,4]triazol-3(5H)-ylidene)urea* (**6e**). Starting from 0.142 g (0.5 mmol) of **3b** and 0.0768 g (0.0639 mL, 0.5 mmol) of 4-chlorophenyl isocyanate; eluent: ethyl acetate:methanol (9:1, v/v); crystallized from methanol; yield 0.105 g (48%); m.p. 256–260 °C; IR (KBr, cm^−1^): 3408, 3374, 3272, 3063, 3037, 2927, 2866, 1680, 1622, 1574, 1492, 1391, 1309, 1286, 1270, 1229, 1151, 825; ^1^H-NMR (300 MHz, DMSO-d_6_): 2.31 (s, 3H, CH_3_), 3.51 (s, 4H, CH_2_-CH_2_), 4.21–4.27 (m, 2H, CH_2_), 4.30–4.36 (m, 2H, CH_2_), 6.17 (br.s, 1H, NH), 7.21–7.25 (m, 4H, Ar), 7.63 (d, *J* = 8.8 Hz, 2H, Ar), 7.93 (d, *J* = 8.3 Hz, 2H, Ar); ^13^C-NMR (75 MHz, DMSO-d_6_): 20.98, 45.28 (two overlapping signals), 50.88 (two overlapping signals), 119.87 (two overlapping signals), 121.37 (two overlapping signals), 124.90, 128.65 (two overlapping signals), 129.46 (two overlapping signals), 135.08, 136.53, 140.62, 147.01, 150.60, 155.08, 158.32; *m/z* (ESI): 437 and 439 [M + H]^+^. Anal. Calcd for C_21_H_21_ClN_8_O (436.90): C, 57.73; H, 4.84; N, 25.65. Found: C, 57.78; H, 4.81; N, 25.58.

*1-(4-Chlorophenyl)-3-(2-(4-chlorophenyl)-7-(4,5-dihydro-1H-imidazol-2-yl)-6,7-dihydro-2H-imidazo[2,1-c][1,2,4]triazol-3(5H)-ylidene)urea* (**6f**). Starting from 0.1519 g (0.5 mmol) of **3c** and 0.0768 g (0.0639 mL, 0.5 mmol) of 4-chlorophenyl isocyanate; eluent: chloroform:methanol (9.5:0.5, v/v); yield 0.12 g (52%); m.p. 256–260 °C; IR (KBr, cm^−1^): 3385, 3291, 3101, 2955, 2927, 2876, 1686, 1662, 1630, 1584, 1514, 1490, 1304, 1234, 1145, 1090, 827; ^1^H-NMR (300 MHz, DMSO-d_6_): 3.51 (s, 4H, CH_2_-CH_2_), 4.20–4.27 (m, 2H, CH_2_), 4.30–4.35 (m, 2H, CH_2_), 6.21 (br.s, 1H, NH), 7.24–7.27 (m, 2H, Ar), 7.46–7.49 (m, 2H, Ar), 7.62–7.65 (m, 2H, Ar), 8.16–8.19 (m, 2H, Ar), 9.36 (s, 1H, NH); ^13^C-NMR (75 MHz, DMSO-d_6_): 45.38 (two overlapping signals), 50.89 (two overlapping signals), 119.94, 122.30, 125.13, 128.71 (two overlapping signals), 128.95 (three overlapping signals), 129.33 (two overlapping signals), 137.87, 140.42, 146.91, 150.80, 154.98, 158.06; *m/z* (ESI): 437 and 439 [M + H]^+^. Anal. Calcd for C_20_H_18_Cl_2_N_8_O (457.32): C, 52.53; H, 3.97; N, 24.50. Found: C, 52.44; H, 3.99; N, 24.45.

*1-(7-(4,5-Dihydro-1H-imidazol-2-yl)-2-phenyl-6,7-dihydro-2H-imidazo[2,1-c][1,2,4]triazol-3(5H)-ylidene)-3-(naphthalen-1-yl)urea* (**6g**). Starting from 0.135 g (0.5 mmol) of **3a** and 0.085 g (0.5 mmol) of 1-naphthyl isocyanate; yield 0.11 g (50%); m.p. 190–194 °C; IR (KBr, cm^−1^): 3367, 3204, 3051, 3012, 2966, 2878, 1678, 1638, 1599, 1578, 1515, 1492, 1455, 1402, 1341, 1256, 1148, 1044, 772; ^1^H-NMR (500 MHz, DMSO-d_6_, recorded at a temperature of 70 °C): 3.55 (s, 4H, CH_2_-CH_2_), 4.27 (t, 2H, CH_2_), 4.38 (t, 2H, CH_2_), 7.19 (t, 1H, Ar), 7.38 (t, 2H, Ar), 7.43–7.50 (m, 3H, Ar), 7.65 (d, *J* = 7.8 Hz, 1H, Ar), 7.76 (d, *J* = 7.3 Hz, 1H, Ar), 7.86–7.89 (m, 1H, Ar), 8.07 (d, *J* = 7.8 Hz, 2H, Ar), 8.13–8.15 (m, 1H, Ar), 8.82 (s, 1H, NH); ^1^H-NMR (500 MHz, DMSO-d_6_+TFA): 3.83 (s, 4H, CH_2_-CH_2_), 4.44–4.46 (m, 2H, CH_2_), 4.55–4.56 (m, 2H, CH_2_), 7.33–7.34 (m, 1H, Ar), 7.44–7.59 (m, 5H, Ar), 7.68 (d, *J* = 8.3 Hz, 1H, Ar), 7.72–7.73 (m, 1H, Ar), 7.88 (d, *J* = 7.8 Hz, 1H, Ar), 8.00–8.08 (m, 3H, Ar), 9.24 (br.s, 2H, 2xNH^+^), 9.35 (br.s, 1H, NH); ^13^C-NMR (125 MHz, DMSO-d_6_+TFA): 44.23 (two overlapping signals), 46.97, 52.37, 123.10, 123.40, 125.30, 125.61, 126.06, 126.30 (two overlapping signals), 126.50, 128.68, 129.79 (two overlapping signals), 130.47, 134.36 (two overlapping signals), 134.55, 137.32, 145.56, 149.98, 153.87, 158.68; *m/z* (ESI): 439 [M + H]^+^. Anal. Calcd for C_24_H_22_N_8_O (438.48): C, 65.74; H, 5.06; N, 25.55. Found: C, 65.69; H, 5.12; N, 25.48.

*N-((7-(4,5-dihydro-1H-imidazol-2-yl)-2-phenyl-6,7-dihydro-2H-imidazo[2,1-c][1,2,4]triazol-3(5H)-ylidene)carbamoyl)-4-methylbenzenesulfonamide* (**6h**). Starting from 0.135 g (0.5 mmol) of **3a** and 0.0986 g (0.0764 mL, 0.5 mmol) of *p*-toluenesulfonyl isocyanate; yield 0.148 g (63%); m.p. 228–232 °C; IR (KBr, cm^−1^): 3386, 3108, 3030, 2953, 2887, 1687, 1641, 1601, 1589, 1567, 1527, 1499, 1443, 1328, 1244, 1134, 1088, 1004, 914, 562; ^1^H-NMR (500 MHz, DMSO-d_6_): 2.35 (s, 3H, CH_3_), 3.50 (s, 4H, CH_2_-CH_2_), 4.13–4.16 (m, 2H, CH_2_), 4.19–4.22 (m, 2H, CH_2_), 7.23 (t, 1H, Ar), 7.32 (d, *J* = 7.8 Hz, 2H, Ar), 7.39 (t, 2H, Ar), 7.74 (d, *J* = 8.3 Hz, 2H, Ar), 8.01 (d, *J* = 8.3 Hz, 2H, Ar), 8.50 (br.s, 2H, 2 x NH); ^1^H-NMR (500 MHz, DMSO-d_6_+TFA): 2.30 (s, 3H, CH_3_), 3.79 (s, 4H, CH_2_-CH_2_), 4.33–4.35 (m, 4H, CH_2_-CH_2_), 7.23–7.27 (m, 3H, Ar), 7.37 (t, 2H, Ar), 7.73 (d, *J* = 8.3 Hz, 2H, Ar), 8.03 (d, *J* = 8.3 Hz, 2H, Ar), 9.15 (br.s, 3H, NH+NH_2_^+^); ^13^C-NMR (125 MHz, DMSO-d_6_+TFA): 21.18, 44.12 (two overlapping signals), 46.35, 52.13, 122.03 (two overlapping signals), 126.91, 127.80 (two overlapping signals), 129.16 (two overlapping signals), 129.68 (two overlapping signals), 138.21, 138.43, 143.76, 147.62, 149.25, 153.89, 155.81; *m*/*z* (ESI): 467 [M + H]^+^. Anal. Calcd for C_21_H_22_N_8_O_3_S (466.52): C, 54.07; H, 4.75; N, 24.02. Found: C, 54.01; H, 4.69; N, 23.88.

*N-((7-(4,5-dihydro-1H-imidazol-2-yl)-2-(p-tolyl)-6,7-dihydro-2H-imidazo[2,1-c][1,2,4]triazol-3(5H)-ylidene)carbamoyl)-4-methylbenzenesulfonamide* (**6i**). Starting from 0.142 g (0.5 mmol) of **3b** and 0.0986 g (0.0764 mL, 0.5 mmol) of *p*-toluenesulfonyl isocyanate; yield 0.15 g (62%); m.p. 229–231 °C; IR (KBr, cm^−1^): 3346, 3063, 2953, 2889, 2808, 1686, 1644, 1617, 1523, 1443, 1363, 1332, 1315, 1159, 1092, 1022, 1012, 813, 754, 666, 585, 547; ^1^H-NMR (500 MHz, DMSO-d_6_): 2.31 (s, 3H, CH_3_), 2.35 (s, 3H, CH_3_), 3.49 (s, 4H, CH_2_-CH_2_), 4.12–4.15 (m, 2H, CH_2_), 4.18–4.22 (m, 2H, CH_2_), 7.18 (d, *J* = 8.3 Hz, 2H, Ar), 7.32 (d, *J* = 8.3 Hz, 2H, Ar), 7.73 (d, *J* = 8.3 Hz, 2H, Ar), 7.87 (d, *J* = 8.3 Hz, 2H, Ar), 8.35 (br.s, 2H, 2xNH); ^1^H-NMR (500 MHz, DMSO-d_6_+TFA): 2.31 (s, 3H, CH_3_), 2.33 (s, 3H, CH_3_), 3.81 (s, 4H, CH_2_-CH_2_), 4.30–4.33 (m, 2H, CH_2_), 4.35–4.39 (m, 2H, CH_2_), 7.21 (d, *J* = 8.5 Hz, 2H, Ar), 7.30 (d, *J* = 7.2 Hz, 2H, Ar), 7.74 (d, *J* = 7.2 Hz, 2H, Ar), 7.92 (d, *J* = 8.5 Hz, 2H, Ar), 15.46 (br.s, 3H, NH+NH_2_^+^); ^13^C-NMR (125 MHz, DMSO-d_6_+TFA): 20.94, 21.43, 44.08, 46.37, 52.23 (two overlapping signals), 121.96 (two overlapping signals), 127.84 (two overlapping signals), 129.70 (two overlapping signals), 129.81 (two overlapping signals), 135.89, 136.58, 138.48, 143.77, 147.53, 149.21, 153.83, 155.73; *m/z* (ESI): 481 [M + H]^+^. Anal. Calcd for C_22_H_24_N_8_O_3_S (480.54): C, 54.99; H, 5.03; N, 23.32. Found: C, 54.84; H, 5.13; N, 23.25.

*1-(7-(4,5-Dihydro-1H-imidazol-2-yl)-2-phenyl-6,7-dihydro-2H-imidazo[2,1-c][1,2,4]triazol-3(5H)-ylidene)-3-phenylthiourea* (**7a**). Starting from 0.135 g (0.5 mmol) of **3a** and 0.0676 g (0.0597 mL, 0.5 mmol) of phenyl isothiocyanate; yield 0.17 g (84%); m.p. 221–223 °C; IR (KBr, cm^−1^): 3296, 3194, 3174, 3103 3023, 2945, 2878, 1676, 1638, 1596, 1558, 1514, 1460, 1421, 1380, 1325, 1285, 1189, 752, 691; ^1^H-NMR (500 MHz, DMSO-d_6_): 3.41–3.51 (m, 4H, CH_2_-CH_2_), 4.41 (s, 4H, CH_2_-CH_2_), 6.17 (s, 1H, NH), 6.97 (t, 1H, Ar), 7.21 (t, 2H, Ar), 7.31 (d, *J* = 7.2 Hz, 1H, Ar), 7.45 (t, 2H, Ar), 7.55 (d, *J* = 7.4 Hz, 2H, Ar), 7.85 (d, *J* = 7.9 Hz, 2H, Ar), 10.08 (s, 1H, NH); ^13^C-NMR (125 MHz, DMSO-d_6_, recorded at a temperature of 40 °C): 44.98, 49.34 (two overlapping signals), 51.85, 121.80 (two overlapping signals), 121.89 (two overlapping signals), 123.58, 127.27, 128.85 (two overlapping signals), 129.56 (two overlapping signals), 138.17, 141.14, 148.57, 151.75, 155.19, 181.54; *m/z* (ESI): 405 [M + H]^+^. Anal. Calcd for C_20_H_20_N_8_S (404.49): C, 59.39; H, 4.98; N, 27.70. Found: C, 59.42; H, 4.89; N, 27.65.

*1-(7-(4,5-Dihydro-1H-imidazol-2-yl)-2-phenyl-6,7-dihydro-2H-imidazo[2,1-c][1,2,4]triazol-3(5H)-ylidene)-3-(p-tolyl)thiourea* (**7b**). Starting from 0.135 g (0.5 mmol) of **3a** and 0.0746 g (0.5 mmol) of *p*-tolyl isothiocyanate; crystallized from methanol; yield 0.12 g (57%); m.p. 212–214 °C; IR (KBr, cm^−1^): 3298, 3199, 3093, 3016, 2941, 2879, 1678, 1634, 1594, 1555, 1512, 1462, 1424, 1375, 1325, 1310, 1284, 1188, 1138, 752; ^1^H-NMR (500 MHz, CDCl_3_): 2.27 (s, 3H, CH_3_), 3.67–3.76 (m, 4H, CH_2_-CH_2_), 4.51–4.70 (m, 4H, CH_2_-CH_2_), 5.51 (br.s, 1H, NH), 6.91–7.02 (m, 2H, Ar), 7.26–7.39 (m, 5H, Ar), 7.77–7.99 (m, 2H, Ar), 8.10 (s, 1H, NH); *m/z* (ESI): 419 [M + H]^+^. Anal. Calcd for C_21_H_22_N_8_S (418.52): C, 60.27; H, 5.30; N, 26.77. Found: C, 60.19; H, 5.28; N, 26.75.

*1-(4-Chlorophenyl)-3-(7-(4,5-dihydro-1H-imidazol-2-yl)-2-phenyl-6,7-dihydro-2H-imidazo[2,1-c][1,2,4]triazol-3(5H)-ylidene)thiourea* (**7c**). Starting from 0.135 g (0.5 mmol) of **3a** and 0.0848 g (0.5 mmol) of 4-chlorophenyl isothiocyanate; yield 0.13 g (59%); m.p. 229–231 °C; IR (KBr, cm^−1^): 3301, 3192, 3163, 3087, 3004, 2947, 2874, 1675, 1641, 1594, 1555, 1511, 1488, 1461, 1430, 1380, 1223, 1300, 1283, 1241, 1186, 1138, 1089, 916, 822, 778, 751, 644; ^1^H-NMR (500 MHz, DMSO-d_6_): 3.53 (s, 4H, CH_2_-CH_2_), 4.36–4.43 (m, 4H, CH_2_-CH_2_), 6.26 (br.s, 1H, NH), 7.24 (d, *J* = 8.8 Hz, 2H, Ar), 7.30 (t, 1H, Ar), 7.45 (t, 2H, Ar), 7.58 (m, 2H, Ar), 7.82 (d, *J* = 7.8 Hz, 2H, Ar), 10.19 (s, 1H, NH); *m*/*z* (ESI): 439 + 440 +441 [M + H]^+^. Anal. Calcd for C_20_H_19_ClN_8_S (438,94): C, 54.73; H, 4.36; N, 25.53. Found: C, 54.67; H, 4.31; N, 25.21.

*1-(7-(4,5-Dihydro-1H-imidazol-2-yl)-2-phenyl-6,7-dihydro-2H-imidazo[2,1-c][1,2,4]triazol-3(5H)-ylidene)-3-(4-nitrophenyl)thiourea* (**7d**). Starting from 0.135 g (0.5 mmol) of **3a** and 0.09 g (0.5 mmol) of 4-nitrophenyl isothiocyanate; yield 0.15 g (67%); m.p. 253–255 °C; IR (KBr, cm^−1^): 3389, 3193, 3140, 3065, 3033, 2867, 1664, 1595, 1567, 1508, 1460, 1432, 1325, 1301, 1244, 1187, 1110, 772; ^1^H-NMR (500 MHz, DMSO-d_6_): 3.53 (s, 4H, CH_2_-CH_2_), 4.41 (s, 4H, CH_2_-CH_2_), 6.28 (br.s, 1H, NH), 7.34 (t, 1H, Ar), 7.48 (t, 2H, Ar), 7.81 (d, *J* = 8.3 Hz, 2H, Ar), 7.85 (d, *J* = 9.3 Hz, 2H, Ar), 8.08 (d, *J* = 9.3 Hz, 2H, Ar), 10.61 (s, 1H, NH); ^1^H-NMR (500 MHz, DMSO-d_6_+TFA): 3.84 (s, 4H, CH_2_-CH_2_), 4.56 (s, 4H, CH_2_-CH_2_), 7.32 (t, 1H, Ar), 7.45 (t, 2H, Ar), 7.83–7.86 (m, 4H, Ar), 8.03 (d, *J* = 9.3 Hz, 2H, Ar), 9.27 (s, 2H, NH+NH^+^), 10.69 (s, 1H, NH); ^13^C-NMR (125 MHz, DMSO-d_6_+TFA): 44.25 (two overlapping signals), 45.77, 53.05, 120.17 (two overlapping signals), 122.61 (two overlapping signals), 124.91 (two overlapping signals), 128.27, 129.66 (two overlapping signals), 137.41, 142.10, 146.78, 148.82, 150.30, 154.14, 182.08; *m/z* (ESI): 450 [M + H]^+^. Anal. Calcd for C_20_H_19_N_9_O_2_S (449.49): C, 53.44; H, 4.26; N, 28.05. Found: C, 53.34; H, 4.21; N, 28.12.

#### 3.1.4. A General Procedure for Preparation of Compounds **8a–c** and **9**

To a stirring solution of compound **3a** or **3b** in anhydrous chloroform (5 mL) appropriate aryl chloride or sulfonyl chloride (sulfonic acid chloride) and anhydrous triethylamine were added (in the mole ratio of 1:2:6). A mixture was heated in an oil bath at 90 °C for 8 h. The progress of the reaction was controlled by TLC. After completion of the reaction the mixture was evaporated under reduced pressure and to the residue, 10 mL of 20% solution of potassium carbonate was added. The mixture was extracted with chloroform (3 × 20 mL). The organic extract was dried with anhydrous magnesium sulfate(VI), filtered, and concentrated under reduced pressure. The crude product was purified on silica gel by preparative thin-layer chromatography (chromatotron) or crystallization. In this manner, the following compounds were obtained.

*4-Methyl-N-(7-(1-(4-methylbenzoyl)-4,5-dihydro-1H-imidazol-2-yl)-2-(p-tolyl)-6,7-dihydro-2H-imidazo[2,1-c][1,2,4]triazol-3(5H)-ylidene)benzamide* (**8a**). Starting from 0.142 g (0.5 mmol) of **3b**, 0.1546 g (1 mmol) of *p*-toluoyl chloride and 0.30357 g (0.418 mL, 3 mmol) of triethylamine; yield 0.14 g (27%); m.p. 245–248 °C; IR (KBr, cm^−1^): 3069, 3026, 2983, 2915, 2865, 1677, 1656, 1606, 1537, 1501, 1465, 1391, 1339, 1312, 1292, 820, 755; ^1^H-NMR (200 MHz, CDCl_3_): 2.30 (s, 3H, CH_3_), 2.39 (s, 3H, CH_3_), 2.45 (s, 3H, CH_3_), 3.81 (t, 2H, CH_2_), 4.03 (t, 2H, CH_2_), 4.54 (s, 4H, CH_2_-CH_2_), 7.05 (d, *J* = 8.4 Hz, 2H, Ar), 7.20 (d, *J* = 8.0 Hz, 2H, Ar), 7.28 (d, *J* = 7.6 Hz, 2H, Ar), 7.64–7.73 (m, 4H, Ar), 8.08 (d, *J* = 8.0 Hz, 2H, Ar); ^13^C-NMR (50 MHz, CDCl_3_): 21.43, 22.02, 22.18, 45.45, 52.09, 52.62, 52.70, 121.68 (two overlapping signals), 129.12 (two overlapping signals), 129.42 (two overlapping signals), 129.58 (two overlapping signals), 129.66 (two overlapping signals), 130.02 (two overlapping signals), 131.85, 135.29, 136.09, 136.39, 141.96, 143.96, 148.12, 149.75, 151.24, 171.46, 172.65; *m*/*z* (ESI): 520 [M + H]^+^. Anal. Calcd for C_30_H_29_N_7_O_2_ (519.60): C, 69.35; H, 5.63; N, 18.87. Found: C, 69.29; H, 5.61; N, 18.92.

*4-Methoxy-N-(7-(1-(4-methoxybenzoyl)-4,5-dihydro-1H-imidazol-2-yl)-2-(p-tolyl)-6,7-dihydro-2H-imidazo[2,1-c][1,2,4]triazol-3(5H)-ylidene)benzamide* (**8b**). Starting from 0.142 g (0.5 mmol) of **3b**, 0.1706 g (1 mmol) of 4-methoxybenzoyl chloride (*p*-anisoyl chloride) and 0.30357 g (0.418 mL, 3 mmol) of triethylamine; yield 0.1 g (36%); m.p. 258–262 °C; IR (KBr, cm^−1^): 3076, 3010, 2964, 2905, 2838, 1672, 1604, 1537, 1505, 1462, 1417, 1401, 1341, 1316, 1291, 1252, 1172, 1160, 1029, 848, 772; ^1^H-NMR (200 MHz, CDCl_3_): 2.30 (s, 3H, CH_3_), 3.81 (t, 2H, CH_2_), 3.84 (s, 3H, OCH_3_), 3.89 (s, 3H, OCH_3_), 4.03 (t, 2H, CH_2_), 4.54 (s, 4H, CH_2_-CH_2_), 6.87–7.06 (m, 6H, Ar), 7.63 (d, *J* = 8.5 Hz, 2H, Ar), 7.80 (d, *J* = 8.8 Hz, 2H, Ar), 8.14 (d, *J* = 8.8 Hz, 2H, Ar); ^13^C-NMR (50 MHz, CDCl_3_): 21.42, 45.49, 52.15, 52.48, 52.78, 55.83, 56.06, 113.60 (two overlapping signals), 114.32 (two overlapping signals), 121.65 (two overlapping signals), 126.64, 129.42 (two overlapping signals), 130.62, 131.84 (two overlapping signals), 131.87 (two overlapping signals), 131.09, 136.39, 147.95, 149.63, 151.59, 162.66, 163.77, 171.14, 172.24; *m/z* (ESI): 552 [M + H]^+^. Anal. Calcd for C_30_H_29_N_7_O_4_ (551.60): C, 65.32; H, 5.30; N, 17.78. Found: C, 65.28; H, 5.34; N, 17.68.

*4-Chloro-N-(7-(1-(4-chlorobenzoyl)-4,5-dihydro-1H-imidazol-2-yl)-2-phenyl-6,7-dihydro-2H-imidazo[2,1-c][1,2,4]triazol-3(5H)-ylidene)benzamide* (**8c**). Starting from 0.135 g (0.5 mmol) of **3a**, 0.175 g (1 mmol) of 4-chlorobenzoyl chloride and 0.30357 g (0.418 mL, 3 mmol) of triethylamine; eluent: dichloromethane:ethyl acetate:2-propanone:triethylamine (3:3:3:1, *v*/*v*/*v*/*v*); yield 0.1 g (18%); m.p. 199–204 °C; IR (KBr, cm^−1^): 3058, 2965, 2926, 2869, 1677, 1638, 1624, 1591, 1534, 1504, 1458, 1398, 1333, 1280, 1089, 1012, 845, 762, 754; ^1^H-NMR (500 MHz, DMSO-*d*_6_): 3.71 (t, 2H, CH_2_), 3.97 (t, 2H, CH_2_), 4.32 (m, 2H, CH_2_), 4.44 (t, 2H, CH_2_), 7.22 (t, 1H, Ar), 7.35 (t, 2H, Ar), 7.49 (d, *J* = 8.8 Hz, 2H, Ar), 7.60 (d, *J* = 8.3 Hz, 2H, Ar), 7.72 (d, *J* = 8.3 Hz, 2H, Ar), 7.77 (d, *J* = 8.3 Hz, 2H, Ar), 8.05 (d, *J* = 8.8 Hz, 2H, Ar); *m/z* (ESI): 547 [M + H]^+^. Anal. Calcd for C_27_H_21_Cl_2_N_7_O_2_ (546.41): C, 59.35; H, 3.87; N, 17.94. Found: C, 59.31; H, 3.84; N, 17.91.

*N-(2-phenyl-7-(1-(phenylsulfonyl)-4,5-dihydro-1H-imidazol-2-yl)-6,7-dihydro-2H-imidazo[2,1-c][1,2,4]triazol-3(5H)-ylidene)benzenesulfonamide* (**9**). Starting from 0.135 g (0.5 mmol) of **3a**, 0.17662 g (0.1276 mL, 1 mmol) of benzenesulfonyl chloride and 0.30357 g (0.418 mL, 3 mmol) of triethylamine; eluent: dichloromethane:ethyl acetate:2-propanone:triethylamine (3:3:3:1, *v*/*v*/*v*/*v*); yield 0.121 g (44%); m.p. 213–216 °C; IR (KBr, cm^−1^): 3058, 2962, 2921, 2873, 1668, 1634, 1598, 1562, 1496, 1447, 1380, 1278, 1174, 1143, 1089, 936, 767, 732, 696, 608; ^1^H-NMR (200 MHz, CDCl_3_): 3.38 (t, 2H, CH_2_), 3.92 (t, 2H, CH_2_), 4.65–4.78 (m, 4H, CH_2_-CH_2_), 7.20–7.38 (m, 3H, Ar), 7.47–7.58 (m, 5H, Ar). 7.64–7.75 (m, 3H, Ar), 7.94–8.00 (m, 4H, Ar); ^13^C-NMR (50 MHz, CDCl_3_); 45.10, 50.17, 51.89, 53.59, 122.23 (two overlapping signals), 126.05 (two overlapping signals), 126.78, 127.64 (two overlapping signals), 128.61 (four overlapping signals), 129.60 (two overlapping signals), 131.70, 134.26, 137.07, 137.35, 143.26, 143.64, 149.10, 150.09; *m*/*z* (ESI): 550 [M + H]^+^. Anal. Calcd for C_25_H_23_N_7_O_4_S_2_ (549.62): C, 54.63; H, 4.22; N, 17.84. Found: C, 54.60; H, 4.26; N, 17.79.

### 3.2. X-ray Crystallography

Diffraction experiments were carried out at room temperature with an Oxford Diffraction SuperNova diffractometer using Cu Kα radiation for **5e** and with an Oxford Diffraction Xcalibur E diffractometer using Mo Kα radiation for **8c**. Diffraction data were processed with CrysAlisPro software [45]. The structures were solved with the program SIR2004 [46] and refined by the full-matrix least-squares method on F2 with SHELXL-2018/3 [47]. Hydrogen atoms were placed in calculated positions and refined as riding on their carriers, except the N-H group H atom in **5e**, which was freely refined. For **8c**, the final difference Fourier map showed a residual electron density peak of ca. 1 e/Å3 close to the inversion center and at a distance of 2.97 Å from O16. This peak was interpreted as a water molecule with occupancy 0.25 disordered around the inversion center. The H atom positions of the disordered water molecule were not determined. Illustrations were prepared with the Mercury software [48].

Crystal data for **5e** (C_20_H_21_N_7_O_2_S, M = 423.50 g/mol): monoclinic, space group P21/c (No. 14), a = 11.0126(11) Å, b = 11.4338(10) Å, c = 15.7222(13) Å, β = 97.819(9)°, V = 1961.3(3) Å3, Z = 4, T = 294 K, μ(Cu Kα) = 1.754 mm^−1^, Dcalc = 1.434 g/cm^3^, 7252 reflections measured (8.104° ≤ 2Θ ≤ 133.18°), 3424 unique (Rint = 0.0271, Rsigma = 0.0346), which were used in all calculations. The final R1 was 0.0415 (I > 2σ(I)), and wR2 was 0.1244 (all data).

Crystal data for **8c** (C_27_H_21_Cl_2_N_7_O_2_⋯0.25H_2_O), M = 550.91 g/mol): triclinic, space group P-1 (No. 2), a = 8.8784(4) Å, b = 9.4988(5) Å, c = 15.3376(9) Å, α = 81.192(4)°, β = 77.160(4)°, γ = 83.356(4)°, V = 1241.78(11) Å3, Z = 2, T = 298 K, μ(Mo Kα) = 0.304 mm^−1^, Dcalc = 1.43 g/cm^3^, 14,261 reflections measured (8.25° ≤ 2Θ ≤ 52.73°), 4889 unique (Rint = 0.0240, Rsigma = 0.0324), which were used in all calculations. The final R1 was 0.0484 (I > 2σ(I)) and wR2 was 0.1289 (all data).

### 3.3. In Vitro Anticancer Activity

All cell culture reagents were purchased from Sigma (Deisenhofen, Germany). Cancer cell lines were obtained from the German Collection of Microorganisms and Cell Cultures (DSMZ, Braunschweig, Germany). The culture medium for cell lines was RPMI-1640 medium containing 2 g/L HCO_3_^−^ and 10% FCS. Cells were incubated in a humid atmosphere of 5% CO_2_ at 37 °C in 75 cm^2^ plastic culture flasks (Sarstedt, Nümbrecht, Germany) and were passaged shortly before becoming confluent. For the cytotoxicity studies, one-hundred microliters of a cell suspension were seeded into 96 well microtiter plates (Sarstedt) at a density of 1000 cells per well except for the LCLC-103H cell line, which was plated out at 250 cells per well. One day after plating, the cells were treated with the test substance at five concentrations per compound. The 1000-fold concentrated stock solutions in DMF or DMSO were serially diluted by 50% in DMF or DMSO to give the feed solutions, which were diluted 500-fold into the culture medium. The controls received DMF or DMSO. Each concentration was tested in eight wells, with each well receiving 100 μL of the medium containing the substance. The concentration ranges were chosen to bracket the expected IC_50_ values as best as possible. Cells were then incubated for 96 h, after which time, the medium was removed and replaced with 1% glutaraldehyde/PBS. Optical density (OD) was measured at λ = 570 nm by the use of a Sunrise plate reader (Anthos 2010, Salzburg, Austria). Corrected T/C values were calculated according to the equation: (T/C)_corr_(%) = (O.D._T_ − O.D._c.0_)/(O.D._C_ − O.D._c.0_) × 100, where O.D.T is the mean absorbance of the treated cells, O.D._C_ the mean absorbance of the controls, and O.D._c.0_ the mean absorbance at the time the drug was added. The IC_50_ values were estimated by linear least-squares regression of the T/C_corr_ values versus the logarithm of the substance concentration; only concentrations that yielded T/C_corr_ values between 10% and 90% were used in the calculation. The reported IC_50_ values are the averages of three independent experiments.

### 3.4. Annexin V Assay

For this assay, the SISO cell line was used. Cells were detached by trypsinization and counted with a Coulter Counter Z2. Two-hundred fifty-thousand cells were seeded in 2 mL per well of a 6 well plate and allowed to attach overnight. The stock solutions of compound **5m** dissolved in DMF were added to the culture medium to the desired end concentration of 10 µM. For the control, only the solvent was added. The old medium was removed, and 3 mL of fresh medium containing the test compound were added to each well. The plates were incubated for 24 h. After centrifugation, supernatants were removed, and the cells were washed once with PBS, then 500 µL of a 25% trypsin/EDTA/PBS solution were added to each well. Plates were incubated for 5 min, and 1.0 mL of medium was added per well. Cells were resuspended, transferred to 1.5 mL tubes and centrifuged for 5 min. The supernatant was discarded, and 500 µL of 1× binding buffer were added to each tube followed by a 5 min centrifugation. Afterward, the supernatant was removed and 50 µL of 1× binding buffer were added to resuspend the cells. Five microliters of Annexin V-FITC staining solution were pipetted into each tube. To obtain a homogenous suspension, tubes were vortexed and incubated in the dark for 15 min at room temperature. Afterward, 500 µL of 1× binding buffer were added per tube to wash the cells. Tubes were centrifuged for 5 min, and the supernatant was aspired completely. The cell pellet was resuspended in 250 µL 1× binding buffer. Immediately before measurement, two-point-five microliters of the PI solution were added. The prepared samples were analyzed by flow cytometry using the FITC signal detector (FL1) and the phycoerythrin emission signal detector (FL2).

## 4. Conclusions

The newly obtained 7-(4,5-dihydro-1*H*-imidazol-2-yl)-2-aryl-6,7-dihydro-2*H*-imidazo[2,1-*c*][1,2,4]triazol-3(5*H*)-imine derivatives **3a**–**c**, **4a**–**e**, **5a**–**n**, **6a**–**i**, **7a**–**d**, **8a**–**c**, and **9** constitute a small library of heterocyclic compounds in the anticancer drugs design process. The tested compounds exhibit cytotoxic activity, and their calculated IC_50_ values are in the range of 2.38–14.74 µM. The most active compounds are amide **4e** and sulfonamide **5l**, whereas compound **5m** shows the highest selectivity for the SISO cell line. Preliminary results from the Annexin-V assay indicate that *N*-(7-(4,5-dihydro-1*H*-imidazol-2-yl)-2-phenyl-6,7-dihydro-2*H*-imidazo[2,1-*c*][1,2,4]triazol-3(5*H*)-ylidene)-4-phenoxybenzenesulfonamide (**5m**) induces apoptosis in human cancer cell line SISO.

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
