# Peer review of "Synthesis, Structure and Cytotoxicity Testing of Novel 7-(4,5-Dihydro-1H-imidazol-2-yl)-2-aryl-6,7-dihydro-2H-imidazo[2,1-c][1,2,4]triazol-3(5H)-Imine Derivatives"

_molecules, 2020, doi:10.3390/molecules25245924_

Round 1
Reviewer 1 Report
Łukasz Balewski and coll describe the synthesis of a series of 7-(4,5-dihydro-1H-imidazol-2-yl)-2-aryl-6,7-dihydro2H-imidazo[2,1-c][1,2,4]triazol-3(5H)-imine derivatives which were tested on human cancer cells to determine the cytotoxic activity.
The compounds are fully caracterized using IR, NMR spectroscopy, mass spectrometry and elementary analysis. Moreover, the crystal structures of tow compounds 5e and 8c were determined by X-ray crystallography.
Although only a few derivatives proved to be cytotoxic agents, and just one of these was submitted to the Annexin-V assay to determine whether it can induce the apoptosis in SISO cell line, in my opinion this article deserves to be published in Molecules, but after the following revisions:
a deeper discussion on SARs should be added;
a comment on 1H NMR and 13C NMR spectra should be added too;
in Supplementary Materials sone 1H NMR spectra (compounds 3a, 3b, 4a, 4b, 4d, 5a-5f, 5h, 5k, 6a, 6c, 6g, 7a, 8a-8c) do not report the ppm values, please add them.
Author Response
Response to Reviewer 1 Comments
Please kindly note that a new figure has been added and the major revised parts are highlighted in red color for the convenience of re-reviewing.
Point 1: a deeper discussion on SARs should be added;
Response 1: A deeper discussion of the structure-activity relationship has been added.
Point 2: a comment on 1H NMR and 13C NMR spectra should be added too;
in Supplementary Materials some 1H NMR spectra (compounds 3a, 3b, 4a, 4b, 4d, 5a-5f, 5h, 5k, 6a, 6c, 6g, 7a, 8a-8c) do not report the ppm values, please add them.
Response 2: A comment on 1H-NMR and 13C-NMR has been added to the text. Supplementary Materials has been revised and ppm values for compounds 3a, 3b, 4a, 4b, 4d, 5a-5f, 5h, 5k, 6a, 6c, 6g, 7a, 8a-8c have been added.

Reviewer 2 Report
This is a well presented paper with a huge amount of synthesis and evaluation. The chemistry is sound and well thought through with unfortunately rather low yielding reactions many of whose final products had to be purified by PLC making their scale-up for a more comprehensive evaluation rather limited.
However, I find the suggested mechanism at fault and this has to be addressed. It appears that 3 moles of the N-chloroimidazole (2) is being used while 2 moles are beeing mentioned in the text. The charges assigned in the middle section do not balance either. I would suggest that in the new mechanism, some locant be assigned to each intermediate and that these be used in the explanatory text to make sense for the reader.
The paper deserves to be published after these few issues are addressed.
Author Response
Response to Reviewer 2 Comments
Please kindly note that a new figure has been added and the major revised parts are highlighted in red color for the convenience of re-reviewing.
Point 1: However, I find the suggested mechanism at fault and this has to be addressed. It appears that 3 moles of the N-chloroimidazole (2) is being used while 2 moles are being mentioned in the text. The charges assigned in the middle section do not balance either. I would suggest that in the new mechanism, some locant be assigned to each intermediate and that these be used in the explanatory text to make sense for the reader. The paper deserves to be published after these few issues are addressed.
Response 1: The reaction mechanism has been revised. A corrected Scheme 1 has been added to the text.

Reviewer 3 Report
Dear Editor, after an in-depth reading of the manuscript (Manuscript ID: molecules-1022109) entitled ‘Synthesis, structure and cytotoxicity testing of novel 7-(4,5-dihydro-1H-imidazole-2-yl)-2-aryl-6,7-dihydro-2H-imidazo[2,1-c][1,2,4]triazol-3(5H)-imine derivative’
the reviewer thinks that it is suitable for publication in Molecules journal, after a substantial revision.
The paper deals with a new imidazo[2,1-c][1,2,4]triazole derivatives endowed with cytotoxic activity evaluated against four human cellular lines: LCLC-103H, SISO, 5637 and RT-112; authors identify compound 5m as the most selective towards the SISO cellular line.
In the reviewer’s opinion, some aspects of the paper have to be modified:
- Abstract: it is too long and reports the synthetic description of the reactions. Authors should highlight the finding of their research.
- Introduction: the description of the imidazoline and triazole scaffolds in derivatives with anticancer activity is well developed but, it would be useful to the reader a chart/figure that reports some of these derivatives. Moreover, some references are old (the 1990s) and some compounds are referred to be in clinical trials, see Nutlins [13, 14] and levamisole [19]: please update the reference or insert the Number Clinical Trial (NCT) or the clinical phase to date.
- Could be useful to the reader understand the type of structures that authors have studied in their research [37-40] to join the current research.
- The authors determined the crystal structures of compounds 5e and 8c: since there are no problems to assign the structures of synthesized compounds, the reviewer thinks that figure 2 and figure 3, could be moved into the supplementary information file as well as the related discussion. The manuscript does not report a molecular modelling study, a study on the solid-state of the compounds for which X-ray crystallography could be very useful. In the last row of the X-ray description, please change C22 with C23.
- For the chosen of the compound 5m as the best compound, the results reported in Table 1 are analyzed. Unfortunately for the LCLC-103H and 5637 cellular lines the most of the results are lacking; thus, in my opinion, it is better to eliminate the two related columns and evaluated the remained data. Furthermore, the authors have to make some SARs, evaluating, for example, the influence of the different group in the R2 position of structure A (with ref. to fig 1 page 2).
Other modifications:
- Please, the author should insert the corresponding paragraphs as author guidelines report:
- Introduction,
- Results and Discussions,
- Experimental Section,
- Conclusions.
- Please also insert the subparagraphs in the experimental section as follows:
3.1. Chemistry: after this subparagraph, the characterization of compounds must follow
3.2. X-ray crystallography
3.3. In vitro anticancer activity
3.4. Annexin V assay
- Figure 1: for structure A, change R2 = Ar with R2 = H
- Change in a bold font all compound numbers.
- Throughout the chemistry section, please delete the subparagraph 2.1.1….2.1.4 because these are a repetition of the experimental section. In this section describe the synthetic pathways referring to the schemes 1-4.
- Scheme 1: starting from compound 1a-c + 2, delete the synthesis of 1a-c.
- Scheme 2 and Scheme 4: please change R1-C(=O)Cl with R1-COCl and also modify the other reagents.
- Scheme 4: in compound 9 is unnecessary explicit R=H, use the benzene ring in the formula.
- Intermediates 1a-c, are known in the literature, so the characterization of these compounds is not necessary. Please delete it in the manuscript or move it in the supplementary materials. Moreover, reference 41 is wrong. Please change it with the corrected reference. In addition, please insert the references of 1a and 1b in the manuscript and in the reference section because they are lacking.
- Intermediate 2 is commercially available, and please remove its synthetic description from the manuscript it is not necessary.
- Please insert a general procedure also for compounds 3a-c.
- Some general considerations in the chemistry experimental section: - please change “arom.” with “Ar” in the proton spectrum; -please insert the following wording before the NMR spectrum: 1H-NMR for proton and 13C-NMR for carbon (ex “1H-NMR (500 MHz, DMSO-d6)”); the number of compounds should be in bold.
Author Response
Response to Reviewer 3 Comments
Please kindly note that a new figure has been added and the major revised parts are highlighted in red color for the convenience of re-reviewing.
Point 1: Abstract: it is too long and reports the synthetic description of the reactions. Authors should highlight the finding of their research.
Response 1: Abstract has been revised and some text has been deleted and changed.
Point 2: Introduction: the description of the imidazoline and triazole scaffolds in derivatives with anticancer activity is well developed but, it would be useful to the reader a chart/figure that reports some of these derivatives. Moreover, some references are old (the 1990s) and some compounds are referred to be in clinical trials, see Nutlins [13,14] and levamisole [19]: please update the reference or insert the Number Clinical Trial (NCT) or the clinical phase to date.
Response 2: The text has been revised and a new Figure (Figure 1 in the revised manuscript) has been added accordingly. The Number Clinical Trials (NCT) and clinical phase for mentioned compound have been added.
Point 3: Could be useful to the reader understand the type of structures that authors have studied in their research [37-40] to join the current research.
Response 3: The active compounds: 2-imino-2H-chromen-3-yl-1,3,5-triazines, 3-(benzoxazol/benzothiazol-2-yl)-2H-chromen-2-imines, 8-chloro-5,5-dioxoimidazo[1,2-b][1,4,2]benzodithiazines, 2-amino-4-(3,5,5-trimethyl-2-pyrazolino)-1,3,5-triazines, copper(II) complexes of 2-substituted benzimidazoles and N-(2-pyridyl)imidazolidin-2-ones(thiones), described in our previous publications (references 28,29,38-41) have been included in the revised text:
Point 4: The authors determined the crystal structures of compounds 5e and 8c: since there are no problems to assign the structures of synthesized compounds, the reviewer thinks that figure 2 and figure 3, could be moved into the supplementary information file as well as the related discussion. The manuscript does not report a molecular modelling study, a study on the solid-state of the compounds for which X-ray crystallography could be very useful. In the last row of the X-ray description, please change C22 with C23.
Response 4: In the course of experimental work, we highlighted difficulties in describing the structure. Spectroscopic spectra and elementary analysis could indicate both the 6-phenyl-6,8,9,10-tetrahydro-2H-diimidazo[2,1-c:2',1'-e][1,2,4,6]tetrazepin-5(3H)-imine I (with 7-membered ring) and 7-(4,5-dihydro-1H-imidazol-2-yl)-2-phenyl-6,7-dihydro-2H-imidazo[2,1-c][1,2,4]triazol-3(5H)-imine II (with 5-membered ring). For this reason, we decided to perform X-ray examination as more reliable, rather than molecular modelling studies.
[structures I and II in pdf]
Point 5: For the chosen of the compound 5m as the best compound, the results reported in Table 1 are analyzed. Unfortunately for the LCLC-103H and 5637 cellular lines the most of the results are lacking; thus, in my opinion, it is better to eliminate the two related columns and evaluated the remained data. Furthermore, the authors have to make some SARs, evaluating, for example, the influence of the different group in the R2 position of structure A (with ref. to fig 1 page 2).
Response 5: The two columns have been removed from the Table 2 and a deeper discussion concerning the SAR has been presented.
Point 6: Other modifications:
Please, the author should insert the corresponding paragraphs as author guidelines report:
- Introduction,
- Results and Discussions,
- Experimental Section,
- Conclusions.
Please also insert the subparagraphs in the experimental section as follows:
3.1. Chemistry: after this subparagraph, the characterization of compounds must follow
3.2. X-ray crystallography
3.3. In vitro anticancer activity
3.4. Annexin V assay
Figure 1: for structure A, change R2 = Ar with R2 = H
Change in a bold font all compound numbers.
Throughout the chemistry section, please delete the subparagraph 2.1.1….2.1.4 because these are a repetition of the experimental section. In this section describe the synthetic pathways referring to the schemes 1-4.
Scheme 1: starting from compound 1a-c + 2, delete the synthesis of 1a-c.
Scheme 2 and Scheme 4: please change R1-C(=O)Cl with R1-COCl and also modify the other reagents.
Scheme 4: in compound 9 is unnecessary explicit R=H, use the benzene ring in the formula.
Intermediates 1a-c, are known in the literature, so the characterization of these compounds is not necessary. Please delete it in the manuscript or move it in the supplementary materials. Moreover, reference 41 is wrong. Please change it with the corrected reference. In addition, please insert the references of 1a and 1b in the manuscript and in the reference section because they are lacking.
Intermediate 2 is commercially available, and please remove its synthetic description from the manuscript it is not necessary.
Please insert a general procedure also for compounds 3a-c.
Some general considerations in the chemistry experimental section: - please change “arom.” with “Ar” in the proton spectrum; -please insert the following wording before the NMR spectrum: 1H-NMR for proton and 13C-NMR for carbon (ex “1H-NMR (500 MHz, DMSO-d6)”); the number of compounds should be in bold.
Response 6: According to the Reviewer’s suggestions the corresponding paragraphs: 1. Introduction, 2. Results and Discussions, 3. Experimental Section, and 4. Conclusions have been added as well as the subparagraphs in the experimental section as follows: 3.1. Chemistry and the characterization of compounds, 3.2. X-ray crystallography, 3.3. In vitro anticancer activity, and 3.4. Annexin V assay. A new figure (Figure 2 in the revised manuscript) has been added with corrected structure A. All compound numbers have been changed in bold. Subparagraphs 2.1.1 - 2.1.4 have been deleted. Schemes 1,2 and 4 have been corrected. Intermediates 1a-c have been deleted from the manuscript and moved to the supplementary materials. Reference 41 has been changed and an appropriate reference describing the compounds 1a-c has been added to the text. Intermediate 2 is not commercially available due to its instability. It must be synthesized and stored as salt with sulfuric acid. The synthetic procedure of compound 2 has been moved to the supplementary section. The general procedure for compounds 3a-c has been added. In the Experimental Section “arom.” has been replaced with ‘Ar’ as well as ‘1H-NMR’ for proton and ‘13C-NMR’ for carbon have been added. The numbers of compounds are given in bold.

Round 2
Reviewer 1 Report
This revised version deserves to be published in Molecules
Reviewer 3 Report
the paper, in this revised version, is now suitable for the journal